# Timesteps meet Bits: Low-Latency, Accurate, & Energy-Efficient Spiking Neural Networks with ANN-to-SNN Conversion

## Abstract

Spiking Neural Networks (SNN) are now demonstrating comparable accuracy to intricate convolutional neural networks (CNN), all while delivering remarkable energy and latency efficiency when deployed on neuromorphic hardware. In particular, ANN-to-SNN conversion has recently gained significant traction in developing deep SNNs with close to state-of-the-art (SOTA) test accuracy on complex image recognition tasks. However, advanced ANN-to-SNN conversion approaches demonstrate that for lossless conversion, the number of SNN time steps must equal the number of quantization steps in the ANN activation function. Reducing the number of time steps significantly increases the conversion error. Moreover, the spiking activity of the SNN, which dominates the compute energy in neuromorphic chips, does not reduce proportionally with the number of time steps. To mitigate the accuracy concern, we propose a novel ANN-to-SNN conversion framework, that incurs an exponentially lower number of time steps compared to that required in the SOTA conversion approaches. Our framework modifies the SNN integrate-and-fire (IF) neuron model with identical complexity and shifts the bias term of each batch normalization (BN) layer in the trained ANN. To mitigate the spiking activity concern, we propose training the source ANN with a fine-grained $\ell_1$ regularizer with surrogate gradients that encourages high spike sparsity in the converted SNN. Our proposed framework thus yields lossless SNNs with *ultra-low latency*, *ultra-low compute energy*, thanks to the ultra-low timesteps and high spike sparsity, and *ultra-high test accuracy*, for example, 73.30% with only 4 time steps on the ImageNet dataset.

## 1 Introduction

Spiking Neural Networks (SNNs) (Maass, 1997) have emerged as an attractive spatio-temporal computing paradigm for a wide range of complex computer vision (CV) tasks (Pfeiffer et al., 2018). SNNs compute and communicate via binary spikes that are typically sparse and require only accumulate operations in their convolutional and linear layers, resulting in significant compute efficiency. However, training deep SNNs has been historically challenging, because the spike activation function in most neuron models in SNNs yields gradients that are zero almost everywhere. While there has been extensive research on surrogate gradients to mitigate this issue (Bellec et al., 2018; Neftci et al., 2019; O'Connor et al., 2018; Wu et al., 2018; Zenke & Ganguli, 2018; Meng et al., 2022a; Xiao et al., 2022), training deep SNNs

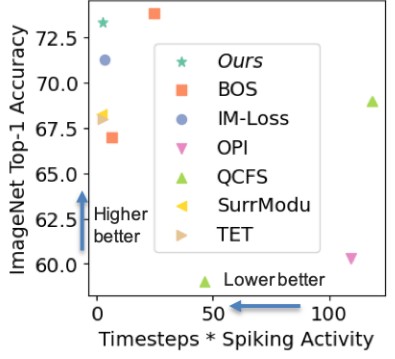

Figure 1: Comparison of the performance-efficiency trade-off between our proposed ANN-to-SNN conversion & other SOTA SNN training methods on ImageNet. Abbreviations are expanded in Section 6.

from scratch is often unable to yield the same accuracies as traditional iso-architecture Artificial Neural Networks (ANN).

ANN-to-SNN conversion, which leverages the advances in SOTA ANN training strategies, has the potential to mitigate this accuracy concern (Sengupta et al., 2019; Rueckauer et al., 2017; Fang et al., 2021). However, since the binary intermediate layer spikes need to be approximated with full-precision ANN activations for accurate conversion, the number of SNN inference time steps required is high. To improve the trade-off between accuracy and time steps, previous research proposed shifting the SNN bias (Deng & Gu, 2021) and initial membrane potential (Bu et al., 2022a; Hao et al., 2023b;a), while leveraging quantization-aware training in the ANN domain (Bengio et al., 2013; Bu et al., 2022b). Although this can eliminate the component of the ANN-to-SNN conversion error incurred by the event-driven binarization, the uneven distribution of the time of arrival of the spikes causes errors, thereby degrading the SNN accuracy. We first uncover that this deviation error is responsible for the accuracy drop in the converted SNNs in ultra-low timesteps. To completely eliminate this deviation as well as other errors with respect to the quantized ANN, we propose a novel conversion framework that completely matches the ANN and SNN activation outputs, while honoring the accumulate-only operation paradigm of SNNs. Our framework yields SNNs with SOTA accuracies among both ANN-to-SNN conversion and backpropagation through time (BPTT) approaches with only $2-4$ time steps. In summary, we make the following contributions.

- We analyze the key sources of error that $(i)$ persist in SOTA ANN-to-SNN conversion approaches, and $(ii)$ degrade the SNN accuracy when using an ultra-low number of time steps.

- We propose a novel ANN-to-SNN conversion framework that exponentially reduces the number of time steps required for SOTA accuracy, eliminates nearly all of the ANN-to-SNN conversion errors, and can be supported in neuromorphic chips, (see Loihi (Davies et al., 2018)).

- We significantly increase the compute efficiency of SNNs by incorporating an additional loss term in our training framework, along with the task-specific loss (e.g., cross-entropy for image recognition). Further, we propose a novel surrogate gradient method to optimize this loss.

Our contributions simultaneously provide ultra-low latency, ultra-high energy efficiency, and SOTA accuracy while surpassing all existing SNN training approaches in performance-efficiency trade-off, where the efficiency is approximated as the product of the number of time steps and the spiking activity, as shown in Fig. 1.

## 2 PRELIMINARIES

### 2.1 ANN & SNN NEURON MODELS

For ANNs used in this work, a block $l$ that takes $a_{l-1}$ as input, consists of a convolution (denoted by $f^{conv}$), batchnorm (denoted by $f^{BN}$), and nonlinear activation (denoted by $f^{act}$), as shown below.

$$a^l = f^{act}(f^{BN}(f^{conv}(a^{l-1}))) = f^{act}(z^l) = f^{act}\left(\gamma^l\left(\frac{W^l a^{l-1} - \mu^l}{\sigma^l}\right) + \beta^l\right), \qquad (1)$$

where $W^l$ denotes the convolutional layer weights, $\mu^l$ and $\sigma^l$ denote the BN running mean and variance, and $\gamma^l$ and $\beta^l$ denote the learnable scale and bias BN parameters. Inspired by (Bu et al., 2022b), we use quantization-clip-floor-shift (QCFS) as the activation function $f^{act}(\cdot)$ defined as

$$a^l = f^{act}(z^l) = \frac{\lambda^l}{L}\text{clip}\left(\left\lfloor \frac{z^l \lambda^l}{L} + \frac{1}{2}\right\rfloor, 0, Q\right), \qquad (2)$$

where $Q$ denotes the number of quantization steps, $\lambda^l$ denotes the trainable QCFS activation output threshold, and $z^l$ denotes the activation input. Note that $\text{clip}(x, 0, \mu) = 0$, if $x < 0$; $x$, if $0 \leq x \leq \mu$; $\mu$, if $x \geq \mu$. QCFS can enable ANN-to-SNN conversion with minimal error for arbitrary $T$ and $Q$, where $T$ denotes the total number of SNN time steps.

The event-driven dynamics of an SNN is typically represented by the IF model where, at each time step denoted as $t$, each neuron integrates the input current $z^l(t)$ from the convolution, followed by BN layer, into its respective state, referred to as membrane potential denoted as $u^l(t)$. The neuron

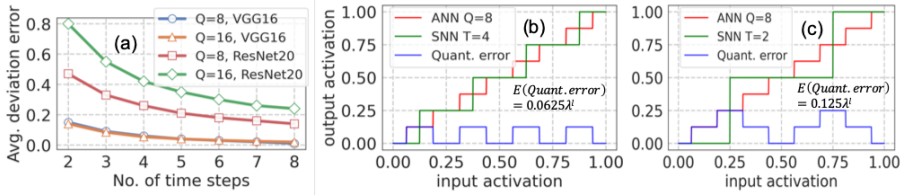

Figure 2: (a) Comparison between the average magnitude of deviation error for different number of time steps with $Q=8$ and $Q=16$, Comparisons of the SNN and ANN output activations, $\phi^l(T)$ and $a^l$ respectively for (b) Q=8 and T=4, (c) Q=8 and T=2. Reducing the number of time steps from 4 to 2 increases the expected quantization error from $0.0625\lambda^l$ to $0.125\lambda^l$.

emits a spike if the membrane potential crosses a threshold value, denoted as $\theta^l$. Assuming $s^{l-1}(t)$ and $s^l(t)$ are the spike inputs and outputs, the IF model dynamics can be represented as

$$u^l(t) = u^l(t-1) + z^l(t) - s^l(t)\theta^l, \tag{3}$$

$$z^l(t) = \left(\gamma^l\left(\frac{W^l s^{l-1}(t)\theta^{l-1} - \mu^l}{\sigma^l}\right) + \beta^l\right), \quad s^l(t) = H(u^l(t-1) + z^l(t) - \theta^l). \tag{4}$$

where $H(\cdot)$ denotes the heaviside function. Note that instead of resetting the membrane potential to zero after the spike firing, we use the reset-by-subtraction scheme where the surplus membrane potential over the firing threshold is preserved and propagated to the subsequent time step.

## 2.2 ANN-TO-SNN CONVERSION

The primary goal of ANN-to-SNN conversion is to approximate the SNN spike firing rate with the multi-bit nonlinear activation output of the ANN with the other trainable parameters being copied from the ANN to the SNN. In particular, rearranging Eq. 3 to isolate the expression for $s^l(t)\theta_l$, summing for $t=1$ to $t=T$, and dividing both sides by $T$, we obtain

$$\frac{\sum_{t=1}^{T} s^l(t)\theta_l}{T} = \frac{\sum_{t=1}^{T} z^l(t)}{T} + \left(-\frac{u^l(0) - u^l(T)}{T}\right). \tag{5}$$

Substituting $\phi^l(T) = \frac{\sum_{t=1}^{T} s^l(t)\theta_l}{T}$ and $Z^l(T) = \frac{\sum_{t=1}^{T} z^l(t)}{T}$ to denote the average spiking rate and presynaptic potential for the layer $l$ respectively, we obtain $\phi^l(T) = Z^l(T) - \left(\frac{u^l(T) - u^l(0)}{T}\right)$. Note that for a very large $T$, $\phi^l(T)$ can be approximated with $Z^l(T)$. Importantly, the resulting function is equivalent to the ANN ReLU activation function, because $\phi^l(T) \geq 0$. However, for the ultra-low $T$ of our use-case, the residual term $\left(\frac{u^l(T) - u^l(0)}{T}\right)$ introduces error in the ANN-to-SNN conversion error, which previous works (Hao et al., 2023a;b; Bu et al., 2022b) refer to as *deviation* error. These works also took into account two other types of conversion errors, namely *quantization* and *clipping* errors. Quantization error occurs due to the discrete nature of $\phi^l(T)$ which has a quantization resolution (QR) of $\frac{\theta^l}{T}$. Clipping error occurs due to the upper bound of $\phi^l(T) = \theta^l$. However, both these errors can be eliminated with the QCFS activation function in the source ANN (see Eq. 2) and setting $\theta^l = \lambda^l$ and $T=Q$. This yields the same QR of $\frac{\theta^l}{T}$ and upper bound of $\theta^l$ as the ANN activation.

## 3 ANALYSIS OF CONVERSION ERRORS

Although we can eliminate the quantization error by setting $T=Q$, the error increases as $T$ is decreased significantly from $Q$ for ultra-low-latency SNNs[1]. This is because the absolute difference between the ANN activations and SNN average post-synaptic potentials increases as $(Q-T)$ increases as shown in Fig 2(b)-(c). Note that $Q$ cannot be too small, otherwise, the source ANN cannot be trained with high accuracy. To mitigate this concern, we propose to improve the SNN

---

[1]Note that $T$ cannot always be equal to $Q$ for practical purposes, since we may require SNNs with different time steps from a single pre-trained ANN

capacity at ultra-low $T$ by embedding the information of both the timing and the binary value of spikes in each membrane potential. As shown later in Section 4, this eliminates the *quantization error* at $T = \log_2 Q$, resulting in an exponential drop in the number of time steps compared to prior works that require $T = Q$ (Bu et al., 2022b). As our work already enables a small value of $T$, the drop in SNN performance with further lower $T < \log_2 Q$ becomes negligible compared to prior works.

Moreover, at ultra-low timesteps, the *deviation error* increases as shown in Fig. 2(a), and even dominates the total error, which implies its importance for our use case. Previous works (Hao et al., 2023b;a) attempted to reduce this error by observing and shifting the membrane potential after some number of time steps, which dictates the upper bound of the total latency. Moreover, (Hao et al., 2023b) require iterative potential correction by injecting or eliminating one spike per neuron at a time, which also increases the inference latency.

That said, the *deviation error* is hard to overcome with the current IF models. To eliminate the deviation error, $v^l(T)$ must fall in the range $[0, \theta_l]$ (Bu et al., 2022b), which cannot be guaranteed without the prior information of the post-synaptic potentials (up to $T$ time steps). The key reason this cannot be guaranteed is the neuron reset mechanism, which dynamically lowers the post-synaptic potential value based on the input spikes. By shifting all neuron resets to the last time step $T$, and matching the ANN activation and SNN post-synaptic values at each time step, *we can completely eliminate* this *deviation error*. This necessitates a new neuron model, and is achieved using our proposed method detailed in Section 4.

## 4  PROPOSED METHOD

In this section, we propose our ANN-to-SNN conversion framework, which involves training the source ANN using the QCFS activation function (Bu et al., 2022b), followed by 1) *shifting the bias term of the BN layers*, and 2) *modifying the IF neuron model* where the neuron spiking mechanism and reset are pushed after the input current accumulation over all the time steps.

### 4.1  ANN-TO-SNN CONVERSION

To enable lossless ANN-to-SNN conversion, the IF layer output should be equal to the bit-wise representation of the output of the corresponding QCFS layer in the $l^{th}$ block, which can be represented as $s^l(t) = a_t^l \; \forall t \in [1, T]$, where $a_t^l$ denotes the $t^{th}$ bit of $a^l$ starting from the most significant bit.

We first show how this is guaranteed for the input block and then for any hidden block $l$ by induction.

**Input Block**: Similar to prior works targeting low-latency SNNs (Bu et al., 2022b;a; Rathi et al., 2020a), we directly use multi-bit inputs that incur multiplications in the first layer, whose overhead is negligible in a deep SNN. Hence, the input to the first IF layer in the SNN (output of the first convolution, followed by BN layer) is identical to the first QCFS layer in the ANN. The first QCFS layer yields the output $a^1$ with $T = \log_2 Q$ bits. The first IF layer also yields identical outputs $s^1(t) = a_t^1$ at the $t^{th}$ time step, with the proposed neuron model as shown later in Eqs. 7 and 8.

**Hidden Block**: To incorporate the information of both the firing time and binary value of the spikes, we multiply the input $s^{l-1}(t)$ of the IF layer (i.e., output of the convolution followed by a BN layer) in the $l^{th}$ block by $2^{(t-1)}$ at the $t^{th}$ time step, which can be easily implemented by a left shifter. This shifting idea is mathematically similar to radix encoding Wang et al. (2022b) and weighted spiking Kim et al. (2018) proposed in previous SNN works. Note that the additional compute overhead due to the shifting is negligible as shown later in Section 6.3. The resulting SNN input current in the $l^{th}$ block is computed as $\hat{z}^l(t) = f^{BN}(f^{conv}(2^{t-1}s^{l-1}(t)))$. The input of the corresponding ANN QCFS layer is $f^{BN}(f^{conv}(a^{l-1}))$ where $a^{l-1}$ can be denoted as $\sum_{t=1}^{T} 2^{t-1}s^{l-1}(t)$ by induction.

*Condition I*: For lossless conversion, let us first satisfy that the accumulated input current over $T$ time steps is equal to the input of the corresponding QCFS layer in the $l^{th}$ block.

Mathematically, representing the composite function $f^{BN}(f^{conv}(\cdot))$ as $g^{ANN}$ and $g^{SNN}$ for the source ANN and its converted SNN respectively, Condition I can be re-written as

$$\sum_{t=1}^{T} g^{SNN}(k \cdot s^{l-1}(t)) = g^{ANN}\left(\sum_{t=1}^{T} k \cdot s^{l-1}(t)\right) \quad \text{where} \quad k = 2^{t-1}. \tag{6}$$

However, this additive property does not hold for any arbitrary source ANN and its converted SNN, due to the BN layer. We satisfy this property by modifying the bias term of each BN layer during the ANN-to-SNN conversion, as shown in Theorem I below, whose proof is in Appendix A.2.

*Theorem I*: For the $l^{th}$ block in the source ANN, let us denote $W^l$ as the weights of the convolutional layer, and $\mu^l$, $\sigma^l$, $\gamma^l$, and $\beta^l$ as the trainable parameters of the BN layer. Let us denote the same parameters of the converted SNN for as $W_c^l$, $\mu_c^l$, $\sigma_c^l$, $\gamma_c^l$, and $\beta_c^l$. Then, Eq. 6 holds true if $W_c^l = W^l$, $\mu_c^l = \mu^l$, $\sigma_c^l = \sigma^l$, $\gamma_c^l = \gamma^l$, and $\beta_c^l = \frac{\beta^l}{T} + (1 - \frac{1}{T})\frac{\gamma^l \mu^l}{\beta^l}$.

*Theorem II*: If Condition I (Eq. 6) is satisfied and the post-synaptic potential accumulation, neuron firing, and reset model adhere to Eqs. 7 and 8 below, the lossless conversion objective i.e., $s^l(t) = a_t^l \ \forall t \in [1, T]$ is satisfied for any hidden block $l$.

$$\hat{z}^l(t) = \left(\gamma_c^l \left(\frac{2^{t-1} W_c^l s^{l-1}(t)\theta^{l-1} - \mu_c^l}{\sigma_c^l}\right) + \beta_c^l\right), \tag{7}$$

$$u^l(1) = \sum_{t=1}^{T} \hat{z}^l(t) + \frac{\theta^l}{2}, \quad s^l(t) = H\left(u^l(t) - \frac{\theta^l}{2^t}\right), \text{ and } u^l(t+1) = u^l(t) - s^l(t)\frac{\theta^l}{2^t}. \tag{8}$$

Note that our neuron model postpones the firing and reset mechanism until after the input current is accumulated from the incoming spikes emitted over all the T time steps in the previous layer, and does not change the complexity of the traditional IF model. Proof of Theorem II is shown in Appendix A.2. Our neuron model can be supported in programmable neuromorphic chips, that implements current accumulation, threshold comparison, and potential reset in a modular fashion. Also, note that our method requires layer-by-layer propagation, as used in advanced conversion works (Hao et al., 2023b;a), since it needs to acquire $\hat{z}^l(T)$, before transmitting the spikes at any time step to the subsequent layer. However, this constraint does not impose any penalty, as layer-by-layer propagation is superior compared to its alternative step-by-step propagation[2] in terms of system efficiency and latency as shown in Appendix A.3.

## 4.2 ACTIVATION SPARSITY

Although our approach explained above can significantly reduce $T$ while eliminating the conversion error, the spiking activity does not reduce proportionally. In fact, there is only a $\sim 3\%$ (36.2% to 33.0%) drop in the spiking activity of a VGG16-based SNN evaluated on CIFAR10 when $T$ decreases from 8 to 4. We hypothesize this is because the SNN tries to pack a similar number of spikes within the few time steps available. To mitigate this concern, we propose a fine-grained regularization method that encourages more zeros in the bit-wise representation of the source ANN. As our approach enforces similarity between the SNN spiking and ANN bit-wise output, this encourages more spike sparsity under ultra low $T$, which in turn, decreases the compute complexity of the SNN when deployed on neuromorphic or sparsity-aware hardware.

The training loss function ($L_{total}$) of our proposed approach is shown below in Eq. 9.

$$L_{total} = L_{CE} + \lambda L_{SP} = L_{CE} + \lambda \sum_{l=1}^{L-1} \sum_{t=1}^{T} \sum_{i=1}^{N} a_t^{i,l}, \tag{9}$$

where $a_t^{i,l}$ denotes the $t^{th}$ bit of the $i^{th}$ activation value in layer $l$, $L_{CE}$ denotes the cross-entropy loss calculated on the softmax output of the last layer $L$, $L_{SP}$ denotes the proposed fine-grained $\ell_1$ regularizer loss, and $\lambda$ is the regularization coefficient. Note that we only accumulate (and do not spike) the post-synaptic potential in the last layer $L$, and hence, we do not incorporate the loss due to $a_t^{i,l}$ for $l=L$. Since $a_t^{i,l} \in \{0, 1\}$, its gradients are either zero or undefined, and so, we cannot directly optimize $L_{SP}$ using backpropagation. To mitigate this issue, inspired by the straight-through estimator (Bengio et al., 2013), we propose a form of surrogate gradient descent as shown below, where $a^{i,l}$ denotes the $t$-bit activation of neuron $i$ in layer $l$:

$$\frac{\partial L_{SP}}{\partial a^{i,l}} = \lambda \sum_{l=1}^{L} \sum_{i=1}^{N} \sum_{t=1}^{T} \frac{\partial a_t^{i,l}}{\partial a^{i,l}}, \quad \text{where} \quad \frac{\partial a_t^{i,l}}{\partial a^{i,l}} = \begin{cases} 1, & \text{if } 0 < a^{i,l} < \lambda^l \\ 0, & \text{otherwise} \end{cases} \tag{10}$$

---

[2] The step-by-step propagation enables the start of processing of the subsequent layer before the processing of the current layer has been finished for all the time steps.

## 5 RELATED WORKS

ANN-to-SNN conversion involves estimating the threshold value in each layer by approximating the activation value of ReLU neurons with the firing rate of spiking neurons (Cao et al., 2015; Rueck-auer et al., 2017; Diehl et al., 2015; Sengupta et al., 2019; Hu et al., 2018). Some conversion works estimated this threshold using heuristic approaches, such as using the maximum (or close to) ANN preactivation value (Rathi et al., 2020b). Others (Kim et al., 2019; Sengupta et al., 2019) proposed weight normalization techniques while setting the threshold to unity. While these approaches helped SNNs achieve competitive classification accuracy on the Imagenet dataset, they required hundreds of time steps for SOTA accuracy. Consequently, there has been a plethora of research (Deng & Gu, 2021; Bu et al., 2022b; Hao et al., 2023b;a) that helped reduce the conversion error while also reducing the number of time steps by an order of magnitude. All these works used trainable thresholds in the ReLU activation function in the ANN and reused the same values for the SNN threshold. In particular, (Deng & Gu, 2021; Li et al., 2021a) proposed a shift in the bias term of the convolutional layers to minimize the conversion error, with the assumption that the ANN and SNN input activations are uniformly and identically distributed. Other works include burst spikes (Park et al., 2019; Li & Zeng, 2022), and signed neuron with memory (Wang et al., 2022a). However, they might not adhere to the bio-plausibility of spiking neurons. Some works also proposed modified ReLU activation functions in the source ANN, including StepReLU (Wang et al., 2023a) and SlipReLU (Jiang et al., 2023) to reduce the conversion error. Lastly, there have been works to address the deviation error in particular. (Bu et al., 2022b) initialized the membrane potential with half of the threshold value to minimize the deviation error; (Hao et al., 2023b;a) rectified the membrane potential after observing its trend for a few time steps; (Meng et al., 2022b) proposed threshold tuning and residual block restructuring.

In contrast to ANN-to-SNN conversion, direct SNN training methods, based on backpropagation through time (BPTT), aim to resolve the discontinuous and non-differentiable nature of the thresholding-based activation function in the IF model. Most of these methods (Lee et al., 2016; Panda & Roy, 2016; Bellec et al., 2018; Neftci et al., 2019; O'Connor et al., 2018; Wu et al., 2018; 2021; Zenke & Ganguli, 2018; Zenke & Vogels, 2021; Meng et al., 2022a; Xiao et al., 2022; Meng et al., 2023; Guo et al., 2022c) replace the spiking neuron functionality with a differentiable model, that can approximate the real gradients (that are zero almost everywhere) with the surrogate gradients. In particular, (Guo et al., 2023a) and (Guo et al., 2022a) proposed a regularizing loss and an information maximization loss respectively to adjust the membrane potential distribution in order to reduce the quantization error due to spikes. Some works optimized the BN layer in the SNN to achieve high performance. For example, (Duan et al., 2022) proposed temporal effective BN, that rescales the presynaptic inputs with different weights at each time-step; (Zheng et al., 2021) proposed threshold-dependent BN; (Kim et al., 2020) proposed batch normalization through time that decouples the BN parameters along the temporal dimension; (Guo et al., 2023b) used an additional BN layer before the spike function to normalize the membrane potential. There have also been works (Rathi et al., 2020a; Datta & Beerel, 2022) where the conversion is performed as an initialization step and is followed by fine-tuning the SNN using BPTT. These hybrid training techniques can help SNNs converge within a few epochs of BPTT while requiring only a few time steps. However, the backpropagation step in these methods requires these gradients to be integrated over all the time steps, which significantly increases the SNN memory footprint during training.

## 6 EXPERIMENTAL RESULTS

In this section, we demonstrate the efficacy of our framework and compare the same with other SOTA SNN training methods for image recognition tasks on CIFAR-10 (Lecun et al., 1998), CIFAR100 (Krizhevsky, 2009), and ImageNet datasets (Deng et al., 2009). Similar to prior works, we evaluate our framework on VGG-16 (Simonyan & Zisserman, 2014), ResNet18 (He et al., 2016), ResNet20, and ResNet34 architectures for the source ANNs. To the best of our knowledge, we are the first to yield ultra-low latency SNNs based on the MobileNetV2 (Sandler et al., 2018) architecture. We compare our method with the SOTA ANN-to-SNN conversion including Rate Norm Layer (RNL) (Ding et al., 2021), Signed Neuron with Memory (SNM) (Wang et al., 2022a), radix encoded SNN (radix-SNN) (Wang et al., 2022b), SNN Conversion with Advanced Pipeline (SNNC-AP) (Li et al., 2021a), Optimized Potential Initialization (OPI) (Bu et al., 2022a), QCFS (Bu et al., 2022b),

Table 1: Comparison of our proposed method to existing ANN-to-SNN conversion approaches on CIFAR10. $Q = 16$ for all architectures, $\lambda=1e-8$. *BOS incurs at least 4 additional time steps to initialize the membrane potential, so their results are reported from $T>4$.

| Architecture | Method | ANN | $T=2$ | $T=4$ | $T=6$ | $T=8$ | $T=16$ | $T=32$ |
|---|---|---|---|---|---|---|---|---|
| VGG16 | RNL | 92.82% | - | - | - | - | 57.90% | 85.40% |
| | SNNC-AP | 95.72% | - | - | - | - | - | 93.71% |
| | OPI | 94.57% | - | - | - | 90.96% | 93.38% | 94.20% |
| | BOS* | 95.51% | - | - | 95.36% | 95.46% | 95.54% | 95.61% |
| | Radix-SNN | - | - | 93.84% | 94.82% | - | - | - |
| | QCFS | 95.52% | 91.18% | 93.96% | 94.70% | 94.95% | 95.40% | 95.54% |
| | **Ours** | 95.82% | 94.21% | 95.82% | 95.79% | 95.82% | 95.84% | 95.81% |
| ResNet18 | OPI | 92.74% | - | - | - | 66.24% | 87.22% | 91.88% |
| | BOS* | 95.64% | - | - | 95.25% | 95.45% | 95.68% | 95.68% |
| | Radix-SNN | - | - | 94.43% | 95.26% | - | - | - |
| | QCFS | 95.64% | 91.75% | 93.83% | 94.79% | 95.04% | 95.56% | 95.67% |
| | **Ours** | 96.68% | 96.12% | 96.68% | 96.65% | 96.67% | 96.73% | 96.70% |
| ResNet20 | OPI | 92.74% | - | - | - | 66.24% | 87.22% | 91.88% |
| | BOS* | 93.3% | - | - | 89.88% | 91.26% | 92.15% | 92.18% |
| | QCFS | 91.77% | 73.20% | 83.75% | 83.79% | 89.55% | 91.62% | 92.24% |
| | **Ours** | 93.60% | 86.9% | 93.60% | 93.57% | 93.66% | 93.75% | 93.82% |

Bridging Offset Spikes (BOS) (Hao et al., 2023b), Residual Membrane Potential (SRP) (Hao et al., 2023a) and direct training methods including Dual Phase (Wang et al., 2023b), Diet-SNN (Rathi et al., 2020a), Information loss minimization (IM-Loss) Guo et al. (2022a), Differentiable Spike Representation (DSR) (Li et al., 2021b), Temporal Efficient Training (Deng et al., 2022), parametric leaky-integrate-and-fire (PLIF) (Fang et al., 2021), RecDis-SNN (Guo et al., 2022c), Membrane Potential Reset (MPR) Guo et al. (2022b), Temporal Effective Batch Normalization (TEBN) (Duan et al., 2022), and Surrogate Module Learning (SML) (Deng et al., 2023). More details about the proposed conversion algorithm and training configurations are provided in Appendix A.1.

## 6.1 EFFICACY OF PROPOSED METHOD

To verify the efficacy of our proposed method, we compare the accuracies obtained by our source ANN and the converted SNN. As shown in Fig. 3, for both VGG and ResNet architectures, the accuracies obtained by our source ANN and converted SNN are identical for $T=log_2Q$. This is expected since we ensure that both the ANN and SNN produce the same activation outputs with the shift of the bias term of each BN layer. Hence, unlike previous works, there is no layer-wise error that gets accumulated and transmitted to the output layer. However, the SNN test ac-

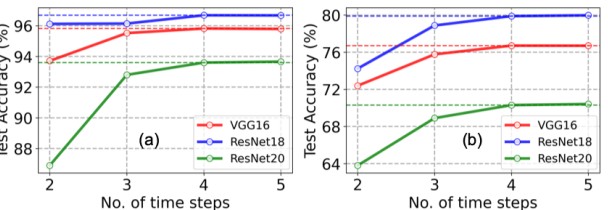

Figure 3: Comparison of the test accuracy of our conversion method for different time steps with $Q = 16$ on (a) CIFAR10 and (b) CIFAR100 datasets. For $T=log_2Q=4$, the ANN & SNN test accuracies are identical. The source ANN accuracies are shown in dotted lines.

curacy starts reducing for lower $T$, which is due to the difference between the ANN and SNN activation outputs but is still higher compared with existing works at the same $T$ as shown below.

## 6.2 COMPARISON WITH SOTA

We compare our proposed framework with the SOTA ANN-to-SNN conversion approaches on CIFAR10 and ImageNet in Table 1 and 2 respectively. For an ultra-low number of time steps, especially $T\leq4$, the test accuracy of the SNNs trained with our method surpasses all the existing methods. Our SNNs can also outperform some of the recently proposed SNNs that incur even higher number of time steps. For example, QCFS reported a test accuracy of $94.95\%$ at $T=8$; our method can surpass that accuracy ($95.82\%$) at $T=4$. Note that (Hao et al., 2023a;b) requires additional time steps to capture the temporal trend of the membrane potential. The authors reported 4 extra time steps for

Table 2: Comparison of our proposed method to existing ANN-to-SNN conversion methods on ImageNet. $Q{=}16$ for both ResNet34 and MobileNetV2, and $\lambda{=}5e{-}10$. *BOS and SRP incurs at least 4 and 8 additional time steps to initialize the membrane potential, so their results are reported from $T{>}4$ and $T{>}8$ respectively.

| Architecture | Method | ANN | $T{=}2$ | $T{=}4$ | $T{=}6$ | $T{=}8$ | $T{=}16$ | $T{=}32$ |
|---|---|---|---|---|---|---|---|---|
| ResNet34 | SNM | 73.18% | - | - | - | - | - | 64.78% |
| | SNNC-AP | 75.36% | - | - | - | - | - | 63.64% |
| | OPI | 93.63% | - | - | - | - | - | 60.30% |
| | BOS* | 74.22% | - | - | 67.12% | 68.86% | 74.17% | 73.95% |
| | SRP* | 74.32% | - | - | - | 57.22% | 67.62% | 68.18% |
| | Radix-SNN | - | - | 72.52% | 73.45% | 73.65% | - | - |
| | QCFS | 74.32% | - | - | - | 35.06% | 59.35% | 69.37% |
| | **Ours** | 75.12% | 54.27% | 75.12% | 75.00% | 75.02% | 75.10% | 75.14% |
| MobileNetV2 | SNNC-AP | 73.40% | - | - | - | - | - | 37.43% |
| | QCFS | 69.02% | 0.20% | 0.26% | 0.53% | 1.12% | 21.74% | 58.45% |
| | **Ours** | 69.02% | 22.62% | 68.81% | 68.89% | 68.98% | 69.02% | 69.01% |

Table 3: Comparison of our method with SOTA BPTT and hybrid training approaches.

| Dataset | Method | Approach | Architecture | Accuracy | Time Steps |
|---|---|---|---|---|---|
| CIFAR10 | Dual-Phase | Hybrid | ResNet18 | 93.27 | 4 |
| | IM-Loss | BPTT | ResNet19 | 95.40 | |
| | MPR | BPTT | ResNet19 | 96.27 | |
| | TET | BPTT | ResNet19 | 94.44 | |
| | RecDis-SNN | BPTT | ResNet19 | 95.53 | |
| | TEBN | BPTT | ResNet19 | 95.58 | |
| | SurrModu | BPTT | ResNet19 | 96.04 | |
| | **Ours** | ANN-to-SNN | ResNet18 | **96.68** | |
| ImageNet | Dspike | Supervised learning | VGG16 | 71.24 | 5 |
| | Diet-SNN | Hybrid | VGG16 | 69.00 | 5 |
| | PLIF | BPTT | ResNet34 | 67.04 | 7 |
| | IM-Loss | BPTT | VGG16 | 70.65 | 5 |
| | RMP-Loss | BPTT | ResNet34 | 65.27 | 4 |
| | SurrModu | BPTT | ResNet34 | 68.25 | 4 |
| | **Ours** | ANN-to-SNN | ResNet34 | **73.30** | 4 |

the accuracy numbers shown in Table 1. As a result, they require at least 5 time steps during inference; their reported accuracies are lower compared to our SNNs at iso-time-step across different architectures and datasets. Moreover, our approach results in >2% increase in test accuracy on both CIFAR10 and ImageNet compared to radix encoding Wang et al. (2022b) (that proposed the similar shifting method we used in this work) for ultra-low time steps (<4), thereby demonstrating the efficacy of our BN bias shift and neuron model. Lastly, as shown in Table 3, our ultra-low-latency accuracies are also higher compared to other SOTA yet memory-expensive SNN training techniques, such as BPTT and hybrid training, at iso-time-step. Moreover, compared to these, our conversion approach leverages standard ANN training with QCFS activation and only requires to change one parameter of each BN layer that is not repeated across time steps, before the SNN inference process.

## 6.3 COMPUTE EFFICIENCY

Our modified IF model incurs the same number of membrane potential update, neuron firing, and reset, compared to the traditional IF model with identical spike sparsity. The only additional overhead is the left shift operation that is performed on each convolutional layer output in each time step. As shown in Table 5 in Appendix A.4, a left shift operation consumes similar energy as an addition operation with identical bit-precision. However, the total number of left shift operations is significantly lower than the number of addition operations incurred in the SNN for the spiking convolution operation. Intuitively, this is because the computational complexity of the spiking convolution operation and the left shift operation is $\mathcal{O}(sk^2 c_{in} c_{out} HW)$ and $\mathcal{O}(c_{out} HW)$ respectively, where $s$ denotes the sparsity. Note that $k$ denotes the kernel size, $c_{in}$ and $c_{out}$ denote the number of input and output channels respectively, and $H$ and $W$ denote the spatial dimensions of the activation map. Even with a sparsity of 90%, for $c_{in}{=}512$ and $k{=}3$, in ResNet18, we have $\frac{sk^2 c_{in} c_{out} HW}{c_{out} HW}{=}406.8$. As shown in

Table 4: Ablation study of the different components of our proposed method on CIFAR10 with VGG16 and ResNet20.

| Architecture | Left shift | BN bias shift | Modified neuron | $T=2$ | $T=4$ | $T=6$ | $T=8$ | $T=16$ |
|---|---|---|---|---|---|---|---|---|
| VGG16 | × | × | × | 91.08% | 93.82% | 94.68% | 94.90% | 95.33% |
| | × | × | ✓ | 92.42% | 94.80% | 95.17% | 95.28% | 95.21% |
| | ✓ | × | × | 93.03% | 95.12% | 95.24% | 95.18% | 95.21% |
| | ✓ | ✓ | × | 93.33% | 95.23% | 95.45% | 95.45% | 95.32% |
| | ✓ | ✓ | ✓ | 94.21% | 95.82% | 95.79% | 95.82% | 95.84% |
| ResNet20 | × | × | × | 71.42% | 83.91% | 84.12% | 88.72% | 92.64% |
| | × | × | ✓ | 76.21% | 90.18% | 91.92% | 92.49% | 92.62% |
| | ✓ | × | × | 76.10% | 91.22% | 91.43% | 92.40% | 92.62% |
| | ✓ | ✓ | × | 79.86% | 91.81% | 92.07% | 93.24% | 93.48% |
| | ✓ | ✓ | ✓ | 86.92% | 93.60% | 93.57% | 93.66% | 93.75% |

Fig. 4(a), the left shifts incur negligible overhead in the total compute energy across both VGG and ResNet architectures. Moreover, left shifts can also be supported in programmable neuromorphic chips, including Loihi (Davies et al., 2021) and Tianjic (Deng et al., 2020).

Our ultra-low-latency SNNs significantly reduce the memory access cost, which is dominated by the successive *read* and *write* operations of the membrane potentials in each time step. Moreover, our fine-grained regularizer significantly reduces the spiking activity of the network. As shown in Fig. 4(b)-(c), with

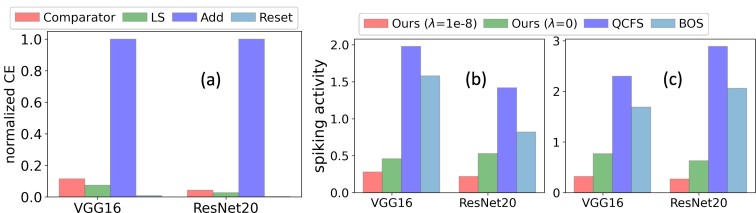

Figure 4: (a) Comparison of the compute energy of each SNN operation with $\lambda=1e-8$ on CIFAR10. Comparison of the spiking activites of the SNNs obtained via our and SOTA conversion methods on (b) CIFAR10 and (c) CIFAR100 with VGG16 and ResNet20. In (a), LS denotes the left shift operation, and CE denotes compute energy.

VGG16, we can obtain a $1.64\times$ reduction for CIFAR10 and $2.40\times$ reduction for CIFAR100. For ResNet-18 on CIFAR10 and ResNet-34 on CIFAR100, the reduction factors are $2.41\times$ and $2.33\times$ respectively. Compared to SOTA conversion approaches (Bu et al., 2022b; Hao et al., 2023b), we obtain $3.73-10.70\times$ reduction in spiking activity. This reduced spiking activity linearly reduces the compute energy. Thus, our proposed ultra-low-latency conversion framework, coupled with high spike sparsity, can significantly reduce the combined system energy consumption. Detailed energy comparisons with ANNs and additional analysis are in Appendix A.4.

## 6.4 ABLATION STUDY OF NEURON MODEL

We conduct ablation studies of our proposed encoding and conversion framework using the traditional IF model. As shown in Table 4, the SNN accuracy drops compared to the ANN counterpart, and the degradation is severe for ultra-low (2-4) time steps. This is due to the deviation error that appears with the normal IF model, and increases significantly at ultra low time steps, dominating the total error. These results validate our hypothesis presented in Section 3. Additionally, when we use the normal IF model, the encoding and bias shift of the BN layers still yield noticeable accuracy increase compared to the QCFS training method that our work is based on, especially for 2-4 time steps. Comparing our results with Table 1, we can also conclude that our conversion framework with the normal IF model yields superior accuracy compared to most of the existing SNN works.

## 7 CONCLUSIONS

In this paper, we first uncover the key sources of error in ANN-to-SNN conversion that have not been completely eliminated in existing works. We propose a novel conversion framework, that completely eliminates all sources of conversion errors when we use the same number of time steps as the bit precision of the source ANN, resulting in an exponential reduction compared to exponential drop compared to existing works. We also propose a fine-grained $\ell_1$ regularizer during the source ANN

training that minimizes the number of spikes in the converted SNN. This significantly increases the compute efficiency, while the ultra-low latency increases the memory efficiency of our SNNs.

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

# A APPENDIX

## A.1 NETWORK CONFIGURATIONS AND HYPERPARAMETERS

We train our source ANNs with average-pooling layers instead of max-pooling as used in prior conversion works (Hao et al., 2023b; Bu et al., 2022b). We also replace the ReLU activation function

in the ANN with QCFS function as shown in Eq. XX, copy the weights from the source ANN to the target SNN and set the QCFS activation threshold $\lambda^l$ equal to the SNN threshold $\theta^l$. Note that $\lambda^l$ is a scalar term for the entire layer to minimize the compute associated with the left-shift of the threshold in the SNN. We set the number of quantization steps $Q$ to 16 for all networks on all datasets.

We leverage the Stochastic Gradient Descent optimizer (Bottou, 2012) with a momentum value of 0.9. We use an initial learning rate of 0.02 for CIFAR-10 and CIFAR-100, and 0.1 for ImageNet, with a cosine decay scheduler (Loshchilov & Hutter, 2017) to lower the learning rate. For CIFAR datasets, we set the value of weight decay to $5 \times 10^{-4}$, while for ImageNet, it is set to $1 \times 10^{-4}$. Additionally, we leverage advanced input augmentation techniques to boost the performance of our source ANN models (DeVries & Taylor, 2017; Cubuk et al., 2019), which can eventually improve the performance of our SNNs. The models for CIFAR datasets are trained for 600 epochs, while those for ImageNet are trained for 300 epochs.

## A.2 PROOF OF THEOREMS & STATEMENTS

*Theorem-I*: For the $l^{th}$ block in the source ANN, let us denote $W^l$ as the weights of the $l^{th}$ hidden convolutional layer, and $\mu^l$, $\sigma^l$, $\gamma^l$, and $\beta^l$ as the trainable parameters of the BN layer. Let us denote the same parameters of the converted SNN for as $W_c^l$, $\mu_c^l$, $\sigma_c^l$, $\gamma_c^l$, and $\beta_c^l$. Then, Eq. 6 holds true if $W_c^l = W^l$, $\mu_c^l = \mu^l$, $\sigma_c^l = \sigma^l$, $\gamma_c^l = \gamma^l$, and $\beta_c^l = \frac{\beta^l}{T} + (1 - \frac{1}{T}) \frac{\gamma^l \mu^l}{\beta^l}$.

*Proof*: Substituting the value of $g^{SNN}$ for the SNN in the left-hand side (LHS) which is equal to the accumulated input current over $T$ time steps, $\sum_{t=1}^{T} \hat{z}_l$, and $g^{ANN}$ in the right-hand side (RHS) of Equation 6, we obtain

$$\sum_{t=1}^{T} \left( \gamma_c^l \left( \frac{2^{t-1} W_c^l s^{l-1}(t) \theta^{l-1} - \mu_c^l}{\sigma_c^l} \right) + \beta_c^l \right) = \left( \gamma^l \left( \frac{\sum_{t=1}^{T} (2^{t-1} W^l s^{l-1}(t) \theta^{l-1}) - \mu^l}{\sigma^l} \right) + \beta^l \right)$$

$$\implies \frac{\gamma_c^l W_c^l \theta^{l-1}}{\sigma_c^l} \sum_{t=1}^{T} 2^{t-1} s^{l-1}(t) + T(\beta_c^l - \frac{\mu_c^l \gamma_c^l}{\sigma_c^l}) = \frac{\gamma^l W^l \theta^{l-1}}{\sigma^l} \sum_{t=1}^{T} 2^{t-1} s^{l-1}(t) + (\beta^l - \frac{\mu^l \gamma^l}{\sigma^l})$$

If we assert $\gamma_c^l = \gamma^l$, $W_c^l = W^l$, $\sigma_c^l = \sigma^l$, the first terms of both LHS and RHS are equal. Substituting $\gamma_c^l = \gamma^l$, $W_c^l = W^l$, and $\sigma_c^l = \sigma^l$ with this assertion, LHS=RHS if their second terms are equal, i.e,

$$T(\beta_c^l - \frac{\mu^l \gamma^l}{\sigma^l}) = (\beta^l - \frac{\mu^l \gamma^l}{\sigma^l}) \implies T\beta_c^l = \beta^l + (T-1)\frac{\mu^l \gamma^l}{\sigma^l} \implies \beta_c^l = \frac{\beta^l}{T} + (1 - \frac{1}{T})\frac{\mu^l \gamma^l}{\sigma^l}$$

*Theorem-II*: If Condition I (Eq. 6) is satisfied and the post-synaptic potential accumulation, neuron firing, and reset model adhere to Eqs. 7 and 8, the lossless conversion objective i.e., $s^l(t) = a_t^l \ \forall t \in [1, T]$ is satisfied for any hidden block $l$.

Repeating Eqs. 7 and 8 here,

$$\hat{z}^l(t) = \left( \gamma^l \left( \frac{2^{t-1} W_c^l s^{l-1}(t) \theta^{l-1} - \mu_c^l}{\sigma_c^l} \right) + \beta_c^l \right), \tag{11}$$

$$u^l(1) = \sum_{t=1}^{T} \hat{z}^l(t), \quad s^l(t) = H\left( u^l(t) - \frac{\theta^l}{2^t} \right), \text{ and } u^l(t+1) = u^l(t) - s^l(t)\frac{\theta^l}{2^t}. \tag{12}$$

Note that $u^l(1) = \sum_{t=1}^{T} \hat{z}^l(t)$ is the original LHS of Eq. 6. Given that Eq. 6 is satisfied due to Theorem-I, we can write $u^l(1) = h^l$, where $h^l$ is the input to the QCFS activation function of the $l^{th}$ block of the ANN. The output of the QCFS function is denoted as $a^l = f^{act}(h^l)$, whose $t^{th}$ bit starting from the most significant bit (MSB) is represented as $a_t^l$. We can check if $a_t^l$ is zero or one, iteratively starting from the MSB, using a binary decision tree approach where we progressively discard one-half of the search range for the subsequent bit checking. With the maximum value of $h^l$ being $\lambda^l$, and $\lambda^l = \theta^l$ (see Section 2.2), $a_1^l = H(h^l - \frac{\theta^l}{2}) = H(u^l(1) - \frac{\theta^l}{2}) = s^l(1)$. To compute $a_2^l$, we can lower $h^l$ by half of the previous range, by first updating $h^l$ as $h^l = h^l - a_1^l \frac{\theta^l}{2}$, and then calculating $a_2^l = H(h^l - \frac{\theta^l}{4}) = H(u^l(2) - \frac{\theta^l}{4})$ which is equal to $s^l(2)$. Similarly, updating $h^l$ to calculate the $t^{th}$ bit $\forall \ t \in [2, T]$ as $h^l = h^l - \frac{\theta^l}{2^{t-1}}$ and then evaluating $a_t^l$ as $a_t^l = H(h^l - \frac{\theta^l}{2^t})$, we obtain $a_t^l = s^l(t), \ \forall t \in [1, T]$.

### A.3 EFFICACY OF LAYER-BY-LAYER PROPAGATION

#### A.3.1 SPATIAL COMPLEXITY

During the SNN inference, the layer-by-layer propagation scheme incurs significantly lower spatial complexity compared to its alternative step-by-step propagation. This is because in step-by-step inference, the computations are localized at a single time step for all the layers, and to process a subsequent time step, all the data, including the outputs and hidden states of all layers at the previous time step, can be discarded. Thus, the spatial inference complexity of the step-by-step propagation is $O(N \cdot L)$, which is not proportional to $T$. In contrast, for layer-by-layer propagation, the computations are localized in a single layer, and to process a subsequent layer, all the data of the previous layers can be discarded. Thus, the spatial inference complexity of the layer-by-layer propagation scheme is $O(N \cdot T)$. Since $T << L$ for deep and ultra low-latency SNNs, the layer-by-layer propagation scheme has lower spatial complexity compared to the step-by-step propagation.

#### A.3.2 LATENCY COMPLEXITY

When operating with step-by-step propagation scheme, let us assume that the $l^{th}$ layer requires $t_s(l)$ to process the input $s^{l-1}(t)$ and yield the output $s^l(t)$. Then, the latency between the input $X$ and the output $s^L(T)$ is $D_{step} = T \sum_{l=1}^{L} t_{step}(l)$.

With layer-by-layer propagation, let us assume that the delay in processing the layer $l$ i.e., outputting the spike outputs for all the time steps ($s^l(t) \ \forall t \in [1, T]$) from the instant the first spike input $s^{l-1}(1)$ is received, is $t_{layer}(l)$. Then, the total latency between the input $X$ and the output $s^L(T)$ is $D_{layer} = \sum_{l=1}^{L} t_{layer}(l)$.

Although each SNN layer is stateful, the computation across the different time steps can be fused into a large CUDA kernel in GPUs when operating with the layer-by-layer propagation scheme (Fang et al., 2020). Even on neuromorphic chips such as Loihi (Davies et al., 2018), there is parallel processing capability. All these imply that $t_{layer}(l) < T \cdot t_{step}(l)$ for any layer $l$. This further implies that $D_{layer} = \sum_{l=1}^{L} t_{layer}(l) < \sum_{l=1}^{L} T \cdot t_{step}(l) < D_{step}$.

In conclusion, the layer-by-layer propagation scheme is generally superior both in terms of spatial and latency complexity compared to the step-by-step propagation, and hence, our method that requires layer-by-layer propagation to operate successfully, does not incur any additional overhead.

### A.4 ENERGY EFFICIENCY DETAILS

Our proposed IF neuron model incurs the same addition, threshold comparison, and potential reset operations as that of a traditional IF model. It simply postpones the comparison and reset operations until after the input current is accumulated over all the $T$ time steps. Thus, our IF model has similar latency and energy complexity compared to the traditional IF model. Moreover, our proposed conversion framework requires that the output of each spiking convolutional layer is left-shifted by $(t-1)$ at the $t^{th}$ time step. However, as shown in Fig. 4, the number of left-shift operations in any network architecture is negligible compared to the total number of addition operations (even with the high sparsity provided by SNNs) incurred in the convolution operation. As a left-shift operation consumes similar energy as an addition operation for both 8-bit and 32-bit fixed point representation as shown in Table 5, the energy overhead of our proposed method is negligible compared to existing SNNs with identical spiking activity. Moreover, the energy overhead due to the addition, comparison and reset operation in our (this holds true for traditional IF models as well) IF model is also negligible compared to the spiking convolution operations as shown in Fig. 4.

Our SNNs yield high sparsity, thanks to our fine-grained $\ell_1$ regularizer, and ultra-low latency, thanks to our conversion framework. While the high sparsity reduces the compute energy compared to existing SNNs, the reduction compared to ANNs is significantly high. This is because ANNs incur multiplication operations in the convolutional layer which is $6.6-31\times$ more expensive compared to the addition operation as shown in Table 5. Thanks to the high sparsity ($71-79\%$) due to the $\ell_1$ regularizer, and the addition-only operations in our SNNs, we can obtain a $7.2-15.1\times$ reduction in the compute energy compared to an iso-architecture SNN, assuming a sparsity of $50\%$ due to the ANN ReLU layers.

Table 5: Comparison of the energy consumed by the different operations in our proposed IF neuron model, and multiplication required in ANNs, on an ASIC (45 nm CMOS technology). Data are obtained from (You et al., 2020; Horowitz, 2014; Gholami et al., 2021; Sekikawa & Yashima, 2023), and our in-house circuit simulations. Note that the reset operation consumes similar energy as addition, and is not shown here.

| Operation | Bit Precision | Energy (pJ) |
|---|---|---|
| Mult. | 32 | 3.1 |
| | 8 | 0.2 |
| Add. | 32 | 0.1 |
| | 8 | 0.03 |
| Left Shift | 32 | 0.13 |
| | 8 | 0.024 |
| Comparator | 32 | 0.08 |
| | 8 | 0.03 |

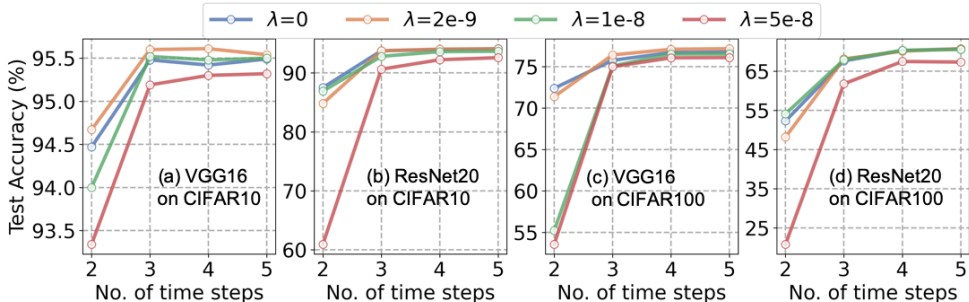

Figure 5: Comparison of the test accuracy of our conversion method for different values of the regularization coefficient $\lambda$.

The memory footprint of the SNNs during inference is primarily dominated by the read and write accesses of the post-synaptic potential at each time step (Datta et al., 2022; Yin et al., 2022). This is because these memory accesses are not influenced by the SNN sparsity since each post-synaptic potential is the sum of $k^2 c_{in}$ weight-modulated spikes. For a typical convolutional layer, $k = 3$, $c_{in} = 128$, and so it is almost impossible that all the $k^2 c_{in}$ spike values are zero for the membrane potential to be kept unchanged at a particular time step[3]. Since our proposed conversion framework significantly reduces the number of time steps compared to previous SNN training methods, it also reduces the number of membrane potential accesses proportionally. Hence, we reduce the memory footprint of the SNN during inference. However, it is hard to accurately quantify the memory savings since that depends on the system architecture and underlying hardware implementation.

## A.5 PERFORMANCE-ENERGY TRADE-OFF WITH BIT-LEVEL REGULARIZER

We can reduce the spiking activity of SNNs using our fine-grained $\ell_1$ regularizer. In particular, by increasing the value of the regularization co-efficient $\lambda$ from 0 to $5e-8$, the spiking activity can be reduced by $2.5-4.1\times$ for different architectures on CIFAR datasets as shown in Fig. 6. However, this comes at the cost of test accuracy, particularly for a very low number of time steps, $T<=3$, as shown in Fig. 5. By carefully tuning the value of $\lambda$, we can obtain SNN models with different sparsity-accuracy trade-offs that can be deployed in scenarios with diverse resource budgets. Using $\lambda=1e-8$ for the CIFAR datasets, and $\lambda=5e-10$ for ImageNet, yields a good trade-off for different time steps. As shown in Fig. 5, $\lambda=1e-8$ yields accuracies that are similar to $\lambda=0$[4] for $T\approx log_2 Q$ for CIFAR datasets. In particular, with ResNet18 for CIFAR10, $\lambda=1e-8$ yields SNN test accuracies

---

[3]Note that the number of weight read and write accesses can be reduced with the spike sparsity, and thus typically do not dominate the memory footprint of the SNN

[4]$\lambda=0$ implies training of the source ANN without our fine-grained regularizer

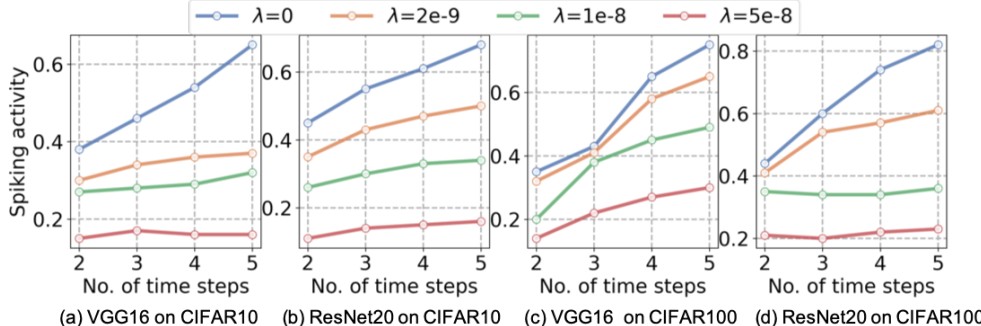

Figure 6: Comparison of the spiking activity of the SNNs obtained via our conversion method for different values of the regularization coefficient $\lambda$.

Table 6: Comparison of our proposed method to existing ANN-to-SNN Conversion approaches on CIFAR100 dataset. $Q$=16 for all architectures, and $\lambda$=$1e-8$. For the reported results below, *BOS incurs 4 additional time steps to initialize the membrane potential, so the total number of time steps is $T$>4.

| Architecture | Method | ANN | $T$=2 | $T$=4 | $T$=6 | $T$=8 | $T$=16 | $T$=32 |
|---|---|---|---|---|---|---|---|---|
| VGG16 | SNM | 74.13% | - | - | - | - | - | 71.80% |
| | SNNC-AP | 77.89% | - | - | - | - | - | 73.55% |
| | OPI | 76.31% | - | - | - | 60.49% | 70.72% | 74.82% |
| | BOS* | 76.28% | - | - | 76.03% | 76.26% | 76.62% | 76.92% |
| | QCFS | 76.28% | 63.79% | 69.62% | 72.50% | 73.96% | 76.24% | 77.01% |
| | **Ours** | 76.71% | 72.39% | 76.71% | 76.74% | 76.70% | 76.78% | 76.82% |
| ResNet20 | OPI | 70.43% | - | - | - | 23.09% | 52.34% | 67.18% |
| | BOS* | 69.97% | - | - | 64.21% | 65.18% | 68.77% | 70.12% |
| | QCFS | 69.94% | 19.96% | 34.14% | 49.20% | 55.37% | 67.33% | 69.82% |
| | **Ours** | 70.30% | 63.80% | 70.30% | 70.33% | 70.45% | 70.49% | 70.52% |

within 0.2% of that of $\lambda = 0$, while reducing the spiking activity by $\sim 2.4\times$ (0.53 to 0.22), which also reduces the compute energy by a similar factor. With ResNet34 for ImageNet, $\lambda = 5e-10$, leads to a 0.4% reduction in test accuracy, while reducing the compute energy by $2\times$. Moreover, as shown in Fig. 6, the spiking activities of our SNNs trained with non-zero values of $\lambda$ do not increase significantly with the number of time steps as that with $\lambda$=0, which also demonstrates the improved compute efficiency resulting from our regularizer.

Table 7: Comparison of our proposed method with SOTA BPTT and hybrid training approaches on CIFAR100 dataset.

| Dataset | Approach | Architecture | Accuracy | Time Steps |
|---|---|---|---|---|
| DSR | BPTT | ResNet18 | 73.35 | 4 |
| Diet-SNN | Hybrid | VGG16 | 69.67 | 5 |
| TEBN | BPTT | ResNet18 | 78.71 | 4 |
| IM-Loss | BPTT | VGG16 | 70.18 | 5 |
| RMP-Loss | BPTT | ResNet19 | 78.28 | 4 |
| SurrModu | BPTT | ResNet18 | 79.49 | 4 |
| **Our Work** | ANN-to-SNN | ResNet18 | **79.89** | 4 |

Table 8: Comparison of the normalized estimated energy of our SNNs on neuromorphic hardware compared to bit-serial accelerators.

| Dataset | Architecture | Neuromorphic | Bit-Serial |
|---------|--------------|--------------|------------|
| CIFAR10 | VGG16 | $1\times$ | $3.57\times$ |
| | ResNet18 | $1\times$ | $4.54\times$ |
| ImageNet | VGG16 | $1\times$ | $3.12\times$ |
| | ResNet34 | $1\times$ | $3.70\times$ |

### A.6 COMPARISON WITH SOTA WORKS FOR CIFAR100

We compare our proposed framework with the SOTA ANN-to-SNN conversion approaches on CIFAR100 in Table 6. Similar to CIFAR10 and ImageNet, for ultra-low number of time steps, especially $T \leq 4$, the test accuracy of our SNN models surpasses existing conversion methods. Moreover, our SNNs can also outperform SOTA-converted SNNs that incur even higher number of time steps. For example, the most recent conversion method, BOSQ reported a test accuracy of $76.03\%$ at $T=6$ (with $4$ time steps added on top of $T = 2$ in Table 6 for the extra $4$ time steps required for potential initialization); our method can surpass that accuracy ($76.71\%$) at $T=4$.

Additionally, as shown in Table 7, our ultra-low-latency accuracies are also higher compared to direct SNN training techniques, including BPTT and hybrid training step at iso-time-step. For example, our method can surpass the test accuracies obtained by the latest BPTT-based SNN training methods (Guo et al., 2023a; Deng et al., 2023) by $0.4-1.6\%$, while significantly reducing the training complexity.

### A.7 COMPARISON WITH BIT-SERIAL QUANTIZATION

Bit-serial quantization is a popular implementation technique for neural network acceleration . It is often desirable for low precision hardware, including in-memory computing chips based on one-bit memory cells such as static random access memory (SRAM) and low-bit cells, such as resistive random access memory (RRAM). Similar to the SNN, it also requires a state variable that stores the intermediate bit-level computations, however, unlike the SNN that compares the membrane state with a threshold at each time step, it performs the non-linear activation function and produces the multi-bit output directly. However, to the best of our knowledge, there is no large-scale bit-serial accelerator chip currently available. Moreover, unlike neuromorphic chips, bit-serial accelerators do not leverage the large activation sparsity demonstrated in our work, and hence, incur significantly higher compute energy compared to neuromorphic chips. Since our SNNs trained with our bit-level regularizer provides a sparsity of $68-78\%$ for different architectures and datasets, they incur $3.1-4.5\times$ lower energy when run on sparsity-aware neuromorphic chips, compared to bit-serial accelerators, as shown in Table 8 .

It can be argued that our approach without our bit-level regularizer leads to results similar to bit-serial computations. However, naively applying bit-serial computing to SNNs with the left-shift approach proposed in this work, would lead to non-trivial accuracy degradations. This is because unlike quantized networks, SNNs can only output binary spikes based on the comparison of the membrane potential against the threshold. Our proposed conversion optimization (bias shift of the BN layers and modification of the IF model) mitigates this accuracy gap, and ensures the SNN computation is identical to the activation-quantized ANN computation. This leads to zero conversion error from the quantized ANNs, and our SNNs achieve identical accuracy with the SOTA quantized ANNs.

### A.8 PSEUDO CODE OF PROPOSED CONVERSION FRAMEWORK

In this section, we summarize our proposed ANN-to-SNN conversion framework in Algorithm 1, which includes training the source ANNs using the QCFS activation function, and then converting to SNNs.

---

**Algorithm 1** : Proposed ANN-to-SNN conversion algorithm

---

1: *Inputs*: ANN model $f^{ANN}(a; W, \mu, \sigma, \beta, \gamma)$ with initial weight $W$, BN layer running mean $\mu$, running variance $\sigma$, learnable scale $\gamma$, and learnable variance $\beta$; Dataset $D$; Quantization step $L$; Initial dynamic thresholds $\lambda$; Learning rate $\epsilon$; Number of SNN time steps $T$

2: *Output*: SNN model $f^{SNN}(a; W, \mu, \sigma, \beta, \gamma)$ & output $s^L(t)$ $\forall t \in [1, T]$ where $L = f^{SNN}$.layers

3: #Source ANN training

4: **for** $e = 1$ to epochs **do**

5:     **for** length of dataset D **do**

6:         Sample minibatch $(a^0, y)$ from D

7:         **for** $l = 1$ to $f^{ANN}$.layers **do**

8:             $a^l = \text{QCFS}(\gamma^l \left( \frac{W^l a^{l-1} - \mu^l}{\sigma^l} \right) + \beta^l)$

9:             $a_t^{i,l} = t^{th}$-bit, starting from MSB, of the $i^{th}$ term in $a^l$

10:         **end for**

11:         loss = CrossEntropy$(a^l, y) + \lambda \sum_{l=1}^{L} \sum_{t=1}^{T} a_t^{i,l}$

12:         **for** $l = 1$ to $f^{ANN}$.layers **do**

13:             $W^l \leftarrow W^l - \epsilon \frac{\partial \text{loss}}{\partial W^l}, \ \mu^l \leftarrow \mu^l - \epsilon \frac{\partial \text{loss}}{\partial \mu^l}, \ \mu^l \leftarrow \sigma^l - \epsilon \frac{\partial \text{loss}}{\partial \sigma^l}$

14:             $\gamma^l \leftarrow \gamma^l - \epsilon \frac{\partial \text{loss}}{\partial \gamma^l}, \ \beta^l \leftarrow \beta^l - \epsilon \frac{\partial \text{loss}}{\partial \beta^l}, \ \lambda^l \leftarrow \lambda^l - \epsilon \frac{\partial \text{loss}}{\partial \lambda^l}$

15:         **end for**

16:     **end for**

17: **end for**

18: #ANN-to-SNN conversion

19: **for** $l = 1$ to $f^{ANN}$.layers **do**

20:     $f^{SNN}.W^l \leftarrow f^{ANN}.W^l, \ f^{SNN}.\theta^l \leftarrow f^{ANN}.\lambda^l, \ f^{SNN}.\mu^l \leftarrow f^{ANN}.\mu^l$

21:     $f^{SNN}.\sigma^l \leftarrow f^{ANN}.\sigma^l, \ f^{SNN}.\gamma^l \leftarrow f^{ANN}.\gamma^l, \ f^{SNN}.\beta^l \leftarrow \frac{f^{ANN}.\beta^l}{T} + (1 - \frac{1}{T}) \frac{f^{ANN}.\gamma^l f^{ANN}.\mu^l}{f^{ANN}.\beta^l}$

22: **end for**

23: #Perform SNN inference on input $a^0$

24: $a^1 = \text{QCFS} \left( f^{SNN}.\gamma^1 \left( \frac{x^0 f^{SNN}.W^1 a^0 - f^{SNN}.\mu^1}{f^{SNN}.\sigma^1} \right) + f^{SNN}.\beta^1 \right)$

25: $s^1(t) = t^{th}$-bit of $a^1$ starting from MSB

26: **for** $l = 2$ to $f^{SNN}$.layers **do**

27:     **for** $t = 1$ to $T$ **do**

28:         $z^l(t) = \left( f^{SNN}.\gamma^l \left( \frac{2^{t-1} f^{SNN}.W^l s^{l-1}(t) - f^{SNN}.\mu^l}{f^{SNN}.\sigma^l} \right) + f^{SNN}.\beta^l \right)$

29:     **end for**

30:     $u^l(1) = \sum_{t=1}^{T} z^l(t) + \frac{\theta^l}{2}$

31:     **for** $t = 1$ to $T$ **do**

32:         $s^l(t) = H(u^l(t) - \frac{f^{SNN}.\theta^l}{2})$

33:         $u^l(t+1) = u^l(t) - s^l(t) \frac{f^{SNN}.\theta^l}{2}$

34:     **end for**

35: **end for**

---

