# OpenReview forum: "Timesteps meet Bits: Low-Latency, Accurate, & Energy-Efficient Spiking Neural Networks with ANN-to-SNN Conversion"
_ICLR.cc/2024/Conference — ICLR 2024 Conference Withdrawn Submission_

### Official Review · Reviewer_JeWJ · 2023-10-17

**Soundness:** 2 fair
**Presentation:** 3 good
**Contribution:** 3 good
**Rating:** 5
**Confidence:** 5

**Summary:**

This paper proposes a novel ANN-SNN framework including using QCFS to get the pretrained ANN and then apply a three-step conversion. This article changes the encoding method of IF neurons to binary and further enhance the representation ability. The paper also proposes a fine-grained regularizer to reduce the spiking activity of the SNN, which further improves the compute efficiency.

**Strengths:**

The paper proposes a novel binary encoded IF neuron and modified the conversion process to fit the binary encoding. The idea of the sparsity regularizer is also interesting.

The experimental results are good. Within 4 timestep, the network can get comparable results with ANN even on ImageNet dataset.

**Weaknesses:**

> “Note that our neuron model simply postpones the firing and reset mechanism until after the input current accumulation over all the T time steps, and does not change the complexity of the traditional.”

I am not sure what this means. Does it indicate that the neuron here will accumulate the input without firing and reset until the last timestep? Authors should explain why they change the neuron behavior. Also, authors should demonstrate more detailed ablation study on the proposed binary encoding and conversion framework using the normal IF neuron.


The sparsity regularizer is a loss function that constrain the activation values in the ANN. However, as the coefficient parameter lambda increases, the performance of SNN will decrease, so the author should discuss more to prove that the advantage of reducing by  adding constraints is greater than the performance loss. Also, in Figure 4, the spiking activity with lambda=1e-8 is larger than the spiking activity with lambda=0. Does that mean the sparsity regularizer does not work?

**Questions:**

Please refer to the concerns addressed in weakness.

---

> ### Author Response · Authors · 2023-11-17
> **Response to Reviewer JeWJ**
>
> Thanks for your valuable comments and suggestions to improve the quality of our work. Please see our response below and the revised version of the manuscript.
>
> **Changing the neuron behavior**
>
> We change the IF neuron model for zero conversion error from the ANN to SNN. Without this change, there will be a deviation error as explained in Section 3, that dominates the total conversion error for ultra-low time-step SNNs (please see Fig 2(a)). With our proposed modifications, we obtain the same accuracy as the ANN as the conversion error has been completely eliminated. Moreover,  our proposed modifications do not change the complexity of the IF model, and can be supported in neuromorphic chips that implement current accumulation, threshold comparison, and potential reset in a modular fashion.
>
> **Ablation study on the proposed binary encoding and conversion framework using the normal IF neuron**
>
> We have now conducted ablation studies of our proposed encoding and conversion framework using the normal IF model. As shown below, the SNN accuracy drops compared to the ANN counterpart, and the degradation is severe for ultra low time steps. This is due to the deviation error that appears with the normal IF model, and increases significantly, dominating the total error at ultra low time steps.
>
>
> | **Architecture** | **Left Shift** | **BN Bias Shift** | **Modified neuron** | **T=2** |**T=4** |**T=6** |**T=8** |**T=16** |
> |------------------|----------------|--------------|------------------|------------------|------------------|------------------|------------------|------------------|
> | VGG16            | No             | No        | No           |  91.08%  |  93.82%  |  94.68%  |  94.90%  |  95.33%  |
> | VGG16            | Yes             | No      | No          | 93.03%  |  95.12%  |  95.24%  |  95.18%  |  95.21%  |
> | VGG16            | Yes               | Yes        | No          |  93.33%  |  95.23%  |  95.45%  |  95.45%  |  95.32%  |
> | VGG16            | Yes             | Yes         | Yes            |  94.21%  |  95.82%  |  95.79%  |  95.82%  |  95.84%  |
> | ResNet20            | No             | No        | No           |  71.42%  |  83.91%  |  84.12%  |  88.72%  |  92.64%  |
> | ResNet20           | Yes             | No      | No          |  76.10%  |  91.22%  |  91.43%  |  92.40%  |  92.62%  |
> | ResNet20           | Yes               | Yes        | No          |  79.86%  |   91.81%  |  92.07%  |  93.24%  | 93.48%  |
> | ResNet20             | Yes             | Yes         | Yes            | 86.92%   |   93.60% |  93.57% |  93.66%  |  93.75%  |
>
>
> However, the encoding and bias shift of the BN layers (with the normal IF model) still yield some accuracy improvement compared to the QCFS training method that our work is based on. These results have been illustrated in the newly created Section 6.4 and Table 4 of the revision. We have also highlighted that our conversion framework with the normal IF model still yields superior accuracy compared to most of the existing SNN works at 2-4 time steps.
>
> **Trade-off between performance and energy with sparsity regularizer**
>
> The proposed sparsity regularizer indeed trades-off accuracy with spiking activity, and the value of the regularizer co-efficient $\lambda$ is carefully tuned such that the advantage of the spiking activity (and thus the compute energy) reduction outweighs the disadvantage of the SNN performance loss.
>
> For relatively easier datasets such as CIFAR10 and CIFAR100, $\lambda=1e{-}8$ yield almost similar accuracies (within $0.3$% accuracy drop) for 4 time steps while the compute energy reduces by $2.4\times$ on average. This may be highly desirable for low-resource edge deployments. For tougher datasets such as ImageNet, we need to choose a smaller value of $\lambda$, otherwise, the accuracy degradation becomes significant. Choosing $\lambda{=}5e{-}10$, leads to a $0.4$% reduction in test accuracy, while reducing the compute energy by $2\times$. Detailed analysis on the efficacy of our regularizer is provided in Appendix A.5. Thus, our proposed sparsity regularizer can yield a range of SNN models with different accuracy-energy trade-offs that can be deployed in scenarios with diverse resource budgets.
>
> **Misplacement of legend**
>
> We have misplaced the legend for $\lambda{=}0$ and $\lambda{=}1e{-}8$, and we sincerely apologize for this mistake. This has been fixed now in the revised version. $\lambda{=}1e{-}8$ indeed gives lower spiking activity compared to $\lambda{=}0$ consistently across different network architectures and datasets. Moreover, Fig. 6 shows that the spiking activity consistently decreases from $\lambda{=}0$ to $\lambda{=}5e{-}8$ for different time steps on CIFAR10.

---

> > ### Comment · Reviewer_JeWJ · 2023-11-19
> > **Response to Authors**
> >
> > Thanks for your comments. However, to me there are still some problems remain unsolved.
> > ### Modifications to IF neurons
> > * I am still confused about how the IF neurons are modified. Please provide detailed description or pseudocode.
> > * If possible, please provide the code.
> >
> > ### Changing the Neuron Behavior
> > * Thanks for authors detailed reponses. I understand that changing the neuron behavior will help eliminate the deviation error. However, authors should provide more evidence to prove that such modification is hardware friendly.
> >
> > ### Ablation Study
> > * I would prefer to see the performance comparison of normal IF neuron and modified neuron without other further techniques. From that we can infer the contribution of the neuron modification to the overall performance improvement. As authors have discussed in the response, since the modified neuron will eliminate the deviation error and lead to a zero-error conversion, the proposed Left Shift and BN Bias Shift techniques will be no longer needed.

---

> ### Author Response · Authors · 2023-11-20
> **Follow-up Response**
>
> Thanks for your prompt reply and we are happy to resolve your concerns. Please see our response below.
>
> **IF Neuron Modification**
>
> The equations governing the traditional IF model for SNNs are
>
> $U_i^{temp}(t)=U_i(t-1)+\sum_j W_{ij}{S_j(t)},  \  \  \  U_i^{temp}(1)=0 $
>
> $S_i(t) = 1 \ \text{if } U_i^{temp}(t)>V^{th} \ \text{else} \ 0$
>
> $U_i(t) = U_i^{temp}(t)-S_i(t)V^{th}$
>
> where $U_i(t)$ denotes the membrane potential of the $i^{th}$ neuron at time step $t$, $S_i(t)$ denotes the binary output of the $i^{th}$ neuron at time step $t$, $V^{th}$ denotes the threshold, and $W_{ij}$ denotes the weight connecting the pre-synaptic neuron $j$ and the neuron $i$. The traditional IF model can not capture the total post-synaptic potentials, up to the total number of time steps. This is due to the neuron reset mechanism, which dynamically lowers the post-synaptic potential value based on the input spikes.
>
> In contrast, our neuron model aims to capture the total synaptic input current without the reset mechanism disturbing the process, which is shown in Eq. (1) below. We then initialize our membrane potential with this current, added to a bias term $\frac{V^{th}}{2}$, which is show in Eq. (2) below. As shown in the QCFS work [1], this bias term helps to minimize the conversion error when the total number of time steps $T$ is not equal to the number of quantization steps in the ANN activation. Lastly, all these, coupled with our neuron firing scheme shown in Eq. (3) and potential reset scheme shown in Eq. (4) contributes in ensuring the average post-activation output of the SNN matches with the ANN counterpart (please see the proof for Theorem-II in Appendix A.2). Note that this modified neuron model is one of the contributing factors (not the only one, the other factors are the proposed bias shift and left-shift) to eliminate the deviation error. More details on this are provided in our response to your next question. The equations governing our modified model are
>
> $U_i^{temp}(t)=U_i^{temp}(t-1)+2^{t{-}1}\sum_j W_{ij}{S_j(t)}, \  \  \  U_i^{temp}(1)=0   \  \  \  \  \  \  \  \   (1) $
>
> $U_i(1) = U_i^{temp}(T)+\frac{V^{th}}{2}  \  \  \  \  \  \  \  \   (2)  $
>
> $S_i(t) = 1 \ \text{if } U_i(t)>\frac{V^{th}}{2^t} \ \text{else} \ 0  \  \  \  \  \  \  \  \   (3) $
>
> $U_i(t) = U_i(t)-S_i(t)\frac{V^{th}}{2^t}  \  \  \  \  \  \  \  \   (4) $
>
> The pseudo-code for this model, representing the same equations above, is shown in the lines 26-33 of the Algorithm-I in Page 20.
>
> **Is the proposed neuron modification hardware-friendly?**
>
> In short, yes, the proposed neuron modification is hardware-friendly. Specifically, as shown in the equations above, our proposed neuron modification simply modifies the order of the neuron reset operation of the IF model, and dynamically right shifts the threshold (please check the division by $2^t$ in Eq. 3 for time step t) at each time step. The traditional neuron model implemented in neuromorphic accelerators typically follows this mode of operation sequentially for each time step : 1) charge accumulation,
> 2) potential comparison,  3) firing (depending on the potential), and 4) reset (depending on the potential). In contrast, our model follows this mode of operation sequentially: 1) charge accumulation for all the time steps, and then for each time step, 2) potential comparison, 3) firing (depending on the potential), and 4) reset (depending on the potential). Since all these four operations are typically implemented as programmable modules in neuromorphic chips [2], our modified neuron model can be easily supported. Lastly, since the threshold is a scalar term for each layer, it requires one right shift operation per layer per time step, which incurs negligible energy as studied in Section 6.3, and is also supported in neuromorphic chips [3]. Lastly, we have confirmed from our collaborators working closely on neuromorphic chip development, that that our neuron model can be supported in Loihi-V2.
>
> [1] T. Bu et al., "Optimal ANN-SNN Conversion for High-accuracy and Ultra-low-latency Spiking Neural Networks", ICLR 2022
>
> [2] [Taking Neuromorphic Computing to the Next Level with Loihi 2 Technology Brief](https://www.intel.com/content/www/us/en/research/neuromorphic-computing-loihi-2-technology-brief.html)
>
> [3] C. Michaelis et al., "Brian2Loihi: An emulator for the neuromorphic chip Loihi using the spiking neural network simulator Brian", Frontiers in Neuroscience, Volume 16 - 2022

---

> > ### Comment · Reviewer_JeWJ · 2023-11-21
> > **Relpy to the Response**
> >
> > > “Note that our neuron model simply postpones the firing and reset mechanism until after the input current accumulation over all the T time steps, and does not change the complexity of the traditional.” **Section 4.1**
> >
> > > In contrast, our neuron model aims to capture the total synaptic input current without the reset mechanism disturbing the process. **Follow-up Response**
> >
> > Thanks for your reply. I can understand the neuron function in the reply. However, I am still confused about the above two sentences. The neuron function provided by the author includes the process of firing and reseting at each time step (**Follow-up Response**, Equation(3)), which is inconsistent with the description "postponing the firing and reset" or "without the reset mechanism". Which one is correct?

---

> > > ### Author Response · Authors · 2023-11-21
> > > **Response**
> > >
> > > Thanks a lot for your continued engagement. We are sorry to confuse you, and we hope our response below can address your existing concerns.
> > >
> > > When we say "we postpone the firing and reset mechanism until after the input current is accumulated over all the T time steps", we mean that unlike the traditional neuron that fires and resets (subject to the potential crossing the threshold) when any input current is received, **our model fires and resets only after the input current is accumulated from all the incoming spikes emitted over the total number of time steps in the previous layer**. We understand our original sentence, that is critical to the description of our neuron model, may confuse the readers, and so, we have re-written this sentence as "Note that our neuron model postpones the firing and reset mechanism until after the input current is accumulated from the incoming spikes emitted over all the T time steps in the previous layer, and does not change the complexity of the traditional IF model."
> > >
> > > When we say "our neuron model aims to capture the total synaptic input current without the reset mechanism disturbing the process", we mean that unlike the traditional IF neuron, our neuron **does not reset during the integration of the synaptic current**. The firing and reset mechanism starts only after the entire input current is accumulated, and then this accumulated current, added with a bias as shown in Eq. (2) below, acts as the initial membrane potential i.e., potential at time step 1. In each time step, starting from time step 1, the neuron fires if the membrane potential crosses the right-shifted threshold as shown in Eq. (3), and the potential resets as shown in Eq. (4).
> > >
> > > $U_i^{temp}(t)=U_i^{temp}(t-1)+2^{t{-}1}\sum_j W_{ij}{S_j(t)}, \  \  \  U_i^{temp}(1)=0   \  \  \  \  \  \  \  \   (1) $
> > >
> > > $U_i(1) = U_i^{temp}(T)+\frac{V^{th}}{2}  \  \  \  \  \  \  \  \   (2)  $
> > >
> > > $S_i(t) = 1 \ \text{if } U_i(t)>\frac{V^{th}}{2^t} \ \text{else} \ 0  \  \  \  \  \  \  \  \   (3) $
> > >
> > > $U_i(t) = U_i(t)-S_i(t)\frac{V^{th}}{2^t}  \  \  \  \  \  \  \  \   (4) $
> > >
> > > We hope this response can resolve your confusion. Please let us know if you are still confused about the operation of our model; we would be happy to answer your questions.

---

> > > > ### Comment · Reviewer_JeWJ · 2023-11-21
> > > > **Relpy to the Response**
> > > >
> > > > Thanks for your reply. Now I understand how modifed neuron works and I would like to express some of my concerns.
> > > >
> > > > * In my opinion, the modified neurons are not as hardware friendly as the authors have claimed in the paper. Here $U_i(1) = U^{temp}_i(T)+V^{th}/2$. Therefore, it will takes at least T time steps for a neuron to transmit spike train to the next neuron, which is inconsistent with common neurons.
> > > >
> > > > * The modified neuron will accumulate all inputs and use the value $V_{th}$ to quantize the output. Such neurons are equivalent to the neurons in activation quantization networks. From a theoretical point of view, the SNN using modified neurons can achieve the same performance as the quantized QCFS pretrained ANN at very low timestep, even if the left shift and bias shift is not adopted. Also, since the SNN with modified neuron is very similar to the quantized networks, why don't we just directly use the more GPU-friendly quantized networks.

---

> ### Author Response · Authors · 2023-11-20
> **Follow-up Response [Contd.]**
>
> **Ablation Study**
>
> We would like to clarify that our modified neuron model can ensure that the average post-activation output of the SNN matches the ANN counterpart (Theorem-II), **provided the inputs also match i.e., the accumulated input current over T time steps is equal to the input of the corresponding QCFS function (Theorem-I)**. The input matching condition is satisfied by the left-shift and BN bias shift together. Thus, both the modified neuron model, and the proposed left-shift + BN bias shift are required to ensure zero conversion error.
>
> Based on your suggestion, we have now included the ablation study of the modified neuron model, without the other techniques presented in this paper, in Table 4, and also below (please see rows 2 and 5). As we can see, the modified neuron model alone cannot ensure the same accuracy as the ANN. This is because the conversion error still exists, as the input to the ANN QCFS activation is not perfectly aligned with the SNN input current. Note that the SNN accuracy with all our operations (please see rows 3 and 6) is identical to the ANN accuracy for $T{=}log_2{Q}{=}4$, where $Q{=}16$ is the number of ANN quantization steps.
>
> | **Architecture** | **Left Shift** | **BN Bias Shift** | **Modified neuron** | **T=2** |**T=4** |**T=6** |**T=8** |**T=16** |
> |------------------|----------------|--------------|------------------|------------------|------------------|------------------|------------------|------------------|
> | VGG16            | No             | No        | No           |  91.08%  |  93.82%  |  94.68%  |  94.90%  |  95.33%  |
> | VGG16            | No             | No      | Yes          | 92.42%  |  94.80%  |  95.17%  |  95.28%  |  95.21%  |
> | VGG16            | Yes             | Yes         | Yes            |  94.21%  |  95.82%  |  95.79%  |  95.82%  |  95.84%  |
> | ResNet20            | No             | No        | No           |  71.42%  |  83.91%  |  84.12%  |  88.72%  |  92.64%  |
> | ResNet20           | No            | No      | Yes       |  76.21%  |  90.18%  |  91.92%  |  92.49%  |  92.62%  |
> | ResNet20             | Yes             | Yes         | Yes            | 86.92%   |   93.60% |  93.57% |  93.66%  |  93.75%  |
>
> Please let us know if our response can resolve your concerns. We would be happy to provide any further supporting explanations and evidence if needed.
>
> **Sharing the code**
>
> We are currently cleaning up our code, and would share the same in a day.

---

> ### Author Response · Authors · 2023-11-21
> **Follow-up Reply to Response**
>
> Thanks for your response again!
>
> **In my opinion, the modified neurons are not as hardware friendly as the authors have claimed in the paper. Here $U_i(1)=U_i^{temp}(T)+\frac{V^{th}}{2}$. Therefore, it will takes at least T time steps for a neuron to transmit spike train to the next neuron, which is inconsistent with common neurons.**
>
> It will take **T time steps of the previous layer** for a neuron to transmit spike to the next layer. However, this does not pose any considerable overhead because the neuron incurs the same number of {charge accumulation, potential comparison, firing, and reset} operations (for the total duration of T time steps) as explained in our response above, and all these operations are typically implemented in a programmable manner in neuromorphic chips. We are slightly changing the order of these operations which has almost no effect (due to their programmability) on the energy and latency of the SNN. The only constraint is that we need to use layer-by-layer propagation scheme, whereas traditional neurons can use either layer-by-layer or step-by-step propagation scheme (both schemes are supported by most neuromorphic chips). However, as shown in Appendix A.3, this constraint does not impose any penalty, as layer-by-layer scheme yields superior latency compared to step-by-step scheme.
>
> **The modified neuron will accumulate all inputs and use the value to quantize the output. Such neurons are equivalent to the neurons in activation quantization networks.**
>
> Please note that neurons in activation quantized networks does not yield 1-bit outputs like SNNs. For the equivalence you mentioned above (the bit-wise representation of the ANN to match the SNN spike train), we need the input and output of the ANN and SNN activation functions to be identical, and for that, we need the left-shifting, BN bias shift, and modification of the neuron, which are all compatible with neuromorphic hardware. All prior works induce errors between the ANN and converted SNN, while our work can completely eliminate the error. We show, for the first time to our best knowledge, it is possible for SNNs to achieve ANN-like accuracies with ultra-low time steps, on neuromorphic hardware. Our outputs are binary and event-driven, not multi-bit like quantized networks.
>
> **From a theoretical point of view, the SNN using modified neurons can achieve the same performance as the quantized QCFS pretrained ANN at very low timestep, even if the left shift and bias shift is not adopted.**
>
> We humbly disagree with this statement. As we show in the proof Theorem-II in Appendix A.2, for the average SNN activation output to match the ANN counterpart with the modified neuron, the prior condition is that the inputs should match too. If the left-shift and bias shift is not adopted the inputs (barring some special conditions) would not match. As a result, the modified neuron alone can not yield the same performance as the quantized QCFS pre-trained ANN. This is also empirically validated in Table 4.
>
> **Since the SNN with modified neuron is very similar to the quantized networks, why don't we just directly use the more GPU-friendly quantized networks.**
>
> This is because the GPU-friendly quantized network can not leverage the activation sparsity compared to neuromorphic hardware. Moreover, the quantized network on GPUs incur multi-bit mutiplication operations for convolutional/linear layers which incur significantly higher energy compared to accumulate operations in neuromorphic hardware. It can be argued that the quantized network may be able to avoid multiplication operations using bit-serial hardware, but then the hardware no longer remains GPU-friendly. Moreover, unlike neuromorphic chips, there are no large-scale bit-serial hardware that we are aware of. Even if we argue that we can build such a bit-serial hardware that can also leverage activation sparsity (at which point the bit-serial hardware will start looking very similar to a neuromorphic hardware), we still need the conversion steps proposed in this work to get zero conversion error from a pre-trained ANN running on GPUs.

---

> > ### Comment · Reviewer_JeWJ · 2023-11-23
> > **Reply to the Response**
> >
> > Thanks for your detailed responses.
> >
> > I have just carefully checked the QCFS paper and the codes they provided. I use the pretrain model in the [ANN_SNN_QCFS](https://github.com/putshua/ANN_SNN_QCFS) and implement the modified neurons as described in this paper based on their codes. For a ResNet-34 model that reported with around 74% ANN accuracy, I evaluated the 8-timestep SNN performance with both normal IF neuron and the modified neuron that mentioned in this paper. Note that here I just changed the neuron into the modified one and kept other settings as default (no left shift).
> > | Neuron     | ANN   | SNN(t=8)|
> > |------------    | -------- |  ---------  |
> > | Normal IF  | 74.28 |   35.09   |
> > | Modified IF| 74.28 |   74.38   |
> > By simply changing the neuron into the modified one and setting the "L" parameter in QCFS code to be the same as the inference time step in SNN, the performance of the model will be almost the same as the origin ANN. This equivalence is not a coincidence, because such equivalence holds true from the theoretical perspective.
> >
> > Furthermore, for the modified neuron, as authors have claimed above, it will indeed take T timesteps of the previous layer for a neuron to transmit spike to the next layer. Then there will be a T-timestep delay at each layer and the latency will accumulate. In other words, the neuron in the last layer will wait at least (L-1)*T timesteps until the first spike sequence arrives, so the modification introduced in the paper will clearly extend the SNN inference time.
> >
> > Authors have mentioned the advantage of activation sparsity of SNNs. However, since the activation value of the origin ANN is often approximated by the firing rate of the converted SNN, there will not be any obvious advantages of converted SNN over origin ANN in terms of activation sparsity.
> >
> > In conclusion, although authors' efforts in responding to my concerns are appreciated, I still want to point out the following issues.
> >
> > * The modification of the neuron brings huge performance increments according to my own experiments. This also proves that the modified neuron will be equivalent to the quantized neuron at a very low certain timestep. Since such SNN will be the very similar to the quantized networks, **authors should provide sufficient evidence about the motivation of converting a quantized network into a "network almost the same as quantized network"**.
> >
> > * Since the modified neurons can significantly improve performance, **this paper should focus more on such modification** instead of introducing such modification in just one sentence and leaving the changed neuron function in the last page of the appendix.
> >
> > * The computation efficiency of the modification neuron need further detailed clarification. As I have discussed above, such modification will servely increase the SNN inference time and the modified neurons in different layers will not be able to work parallelly. **It is still doubtful whether this neuron is hardware friendly.**

---

### Official Review · Reviewer_yJuh · 2023-10-31

**Soundness:** 2 fair
**Presentation:** 3 good
**Contribution:** 2 fair
**Rating:** 5
**Confidence:** 5

**Summary:**

This paper presents a Spiking Neural Network (SNN) architecture with low latency and high model accuracy by introducing an approach of shifting input values by 1 bit (or equivalent to multiplying by 2) at each time step. This shifting mechanism takes advantage of the binary nature of spikes in SNNs, resulting in a significant reduction in the spiking length, scaling down the representation to a logarithmic scale.

**Strengths:**

The paper is written with remarkable clarity, and the experiments are very comprehensive.

**Weaknesses:**

The novelty of this paper may be questionable, as a previously published paper [1] discusses the use of similar shifting technology for converting Artificial Neural Networks (ANN) to SNN. Their work provides a more general theory, demonstrating that the scale of shifting can vary not only to 2 but also to other fractions such as 3, 4, and beyond. Thus, it can be argued that this paper can be considered a specific case or an application of the more comprehensive theory presented in [1]. Alternatively, another published paper by Kim et al. [2] explores a similar idea but approaches it from the opposite direction in shifting.


[1] Wang, Z., Gu, X., Goh, R. S. M., Zhou, J. T., & Luo, T. (2022). Efficient spiking neural networks with radix encoding. IEEE Transactions on Neural Networks and Learning Systems.

[2] Kim, J., Kim, H., Huh, S., Lee, J., & Choi, K. (2018). Deep neural networks with weighted spikes. Neurocomputing, 311, 373-386.


The primary distinction between this paper and the previous work [1] is the object of shifting, with this paper proposing to shift the input left, while [1] focuses on shifting the output right. However, it can be argued that this contribution may not be substantial enough to warrant a separate paper submission to a prestigious conference like ICLR, as these two shifting policies are mathematically equivalent and yield nearly identical results.

In conclusion, while the paper presents an interesting concept for improving SNN performance, its novelty is questionable, given the existence of related work [1] that provides a more general theory and [2] that explores similar ideas from a different perspective. The contribution of changing the direction of shifting alone may not be sufficient to justify a separate publication in ICLR.

**Questions:**

It would be beneficial if the authors could provide a clarification regarding the unique contributions of this paper in relation to the previously mentioned work.

---

> ### Author Response · Authors · 2023-11-17
> **Response to Reviewer yJuh**
>
> Thanks for your insightful review. Please find our response below.
>
> **Comparison with previously proposed radix encoded SNNs [1] and weighted spikes [2]**
>
> Thanks for pointing out the two references, and we apologize for missing them. Upon carefully examining the two papers, we would like to highlight that our conversion framework involves some novel contributions that are not present in the two references. We agree that our shifting technology, albeit technically different, is mathematically similar to that of [1], and in the revised version, we have appropriately positioned our approach against both [1] and [2]. We have also added the comparison of the accuracy of our approach with [1] in Table 1; our approach demonstrates a 2-2.5% increase in test accuracy on CIFAR10 and ImageNet datasets. Moreover, we also demonstrate with our FPGA simulations that the energy incurred in the left shift is insignificant compared to the accumulate operations in the SNN, and this further bolsters the importance of the bit-shifting idea for ultra-low latency SNNs.
>
> The key novel contribution of our paper is demonstrating that our bias shift of the BN layers, and modifications of the IF neuron model are necessary to align the average input and output of the ANN and SNN activations, and ensure zero conversion error from ANNs. These are not proposed in [1] or [2], and thus they still incur conversion losses. While [1] proposes weight and bias shifts of the convolutional layer, that is only for normalization of the input activations, and can not match the ANN and SNN activation outputs. In other words, applying left-shift to the input spikes, and not correcting the biases and IF neuron models, still lead to accuracy loss in the SNN. This is reflected in the 2-2.5% higher test accuracy of our SNNs compared to [1] in Tables 1 and 2 for ultra-low time steps. Furthermore, based on the suggestion of Reviewer **JeWJ**, we have also conducted ablation studies to evaluate the  impact of each operation in our conversion method. As shown in the newly created Table 4, the BN bias shift and the modified IF neuron model (without the left shift) lead to a 1.79% (93.60% vs 91.81%) and 7.04% (86.9% vs 79.86%) increase in test accuracy with ResNet20 on CIFAR10 for 2 and 4 time steps respectively. Both these results demonstrate the efficacy of our novel contributions.
>
> Additionally, we propose a bit-level L1 regularizer that significantly reduces the spiking activity of the SNN compared to existing works. Thus, our proposed approach simultaneously yields SOTA accuracies identical to activation quantized ANNs, and ultra-low energy due to the regularizer. This was also not proposed in [1] or [2], and hence, they would potentially incur up to 2.5x higher energy consumption compared to this work, as shown in Fig. 4(b) and 4(c).
>
> Lastly, in this work, we uncover the conversion errors in existing ANN-to-SNN conversion works, and show that the deviation error is the key bottleneck in yielding ultra low-time-step SNNs with SOTA test accuracy. In our perspective, this is also a novel contribution.
>
> To our best knowledge, this is the first work to yield SNNs with accuracies similar to SOTA ANNs, ultra-low latencies (equal to 2-4 time steps), and lowest compute complexity, simultaneously. Hence, we believe this work is important and timely for the SNN community.
>
> [1] Wang, Z., Gu, X., Goh, R. S. M., Zhou, J. T., & Luo, T. (2022). Efficient spiking neural networks with radix encoding. IEEE Transactions on Neural Networks and Learning Systems.
>
> [2] Kim, J., Kim, H., Huh, S., Lee, J., & Choi, K. (2018). Deep neural networks with weighted spikes. Neurocomputing, 311, 373-386.

---

> > ### Comment · Reviewer_yJuh · 2023-11-22
> >
> > Thanks for the clarification. Although the overall novelty is still limited, the proposed tuning BN bias shift looks interesting and could further improve the model accuracy. Therefore, I raised my rating.

---

> ### Author Response · Authors · 2023-11-22
>
> Thanks very much for increasing your rating. Please note that **we not only propose the BN bias shift, but also a modified neuron model that can be supported in most neuromorphic chips**. These two modifications, along with the left-shift, ensure **absolute zero conversion error of the SNN (no previous work has demonstrated this)** from the pre-trained ANN.
>
> As mentioned in our last response, **this is the first work to yield SNNs with accuracies similar to SOTA ANNs, ultra-low latencies (equal to 2-4 time steps), and lowest compute complexity, simultaneously**. Hence, we believe this work is useful to the SNN community and worthy of presentation at ICLR. However, we respect your decision and rating too.

---

### Official Review · Reviewer_bagt · 2023-11-01

**Soundness:** 2 fair
**Presentation:** 3 good
**Contribution:** 2 fair
**Rating:** 3
**Confidence:** 4

**Summary:**

This paper presents an approach to convert artificial neural networks (ANNs) to spiking neural networks. The method relies on the quantization of activations. It has been shown that the proposed method can improve the accuracy performance of SNNs and reduce their spiking activities.

**Strengths:**

The method improves the accuracy performance of SNNs on ImageNet dataset.
The spiking activities have been reduced.

**Weaknesses:**

The proposed method is similar to quantized neural networks. As such, a comparison is required with quantized neural networks in terms of both accuracy and efficiency.

The type of each vector/matrix (e.g., integer, real, etc) needs to be specified.

**Questions:**

My main question is the difference between this work and bit-serial quantized networks in terms of computations?

---

> ### Author Response · Authors · 2023-11-17
> **Response to Reviewer bagt**
>
> Thanks for your valuable comments. Please see our response below.
>
> **Type of vector/matrix used**
>
> We are not entirely sure of the question, and please let us know if we misunderstand. If you mean the weight matrix, we assume it is fixed-point in this work, as typical in SNN setups. Thus, our proposed approach, similar to other SNN works, requires real-valued accumulate operations. However, the weight precision can be easily reduced to 8-bit integers even without any fancy quantization-aware training tricks, without a significant drop in accuracy, and as a result, our approach can also enable cheaper integer accumulate operations.
>
> **Similarity with bit-serial quantized networks**
>
> Bit-serial quantization is an implementation technique for neural networks, they are not a class of neural networks. Previous works [1, 2] proposed bit-serial quantization to accelerate NNs, and it is often desirable for low precision hardware, including in-memory computing chips based on one-bit memory cells such as static random access memory (SRAM) and low-bit cells, such as resistive random access memory (RRAM). Similar to the SNN, it also requires a state variable that stores the intermediate bit-level computations, however, unlike the SNN that compares the membrane state with a threshold at each time step, it performs the non-linear activation function and produces the multi-bit output directly. However, to the best of our knowledge, there is no large-scale bit-serial accelerator chip currently available. Moreover, unlike neuromorphic chips, bit-serial accelerators do not leverage the large activation sparsity demonstrated in our work, and hence, incur significantly higher compute energy compared to neuromorphic chips. Based on our analytical estimations detailed in Appendix A.5 and tabulated below, **our SNNs incur 3.1-4.5x lower energy when run on sparsity-aware neuromorphic chips, compared to bit-serial accelerators**.
>
> | **Dataset** | **Architecture** | **Neuromorphic Energy** | **Bit-Serial Energy** |
> |------------------|----------------|--------------|------------------|
> | CIFAR10          | VGG16               |1x        | 3.57x           |
> | CIFAR10            | ResNet18              | 1x       | 4.54x            |
> | ImageNet          | VGG16              | 1x      | 3.12x           |
> | ImageNet            | ResNet34            | 1x        | 3.70x          |
>
> We agree that it can be argued our approach **without our bit-level regularizer** leads to results similar to bit-serial computations. However, **naively applying bit-serial computing to SNNs with the left-shift approach proposed in this work, would lead to non-trivial accuracy degradations** as shown in the newly created Table 4 of the revision. This is because unlike quantized networks, SNNs can only output binary spikes based on the comparison of the membrane potential against the threshold. Our proposed conversion optimization, consisting of the bias shift of the BN layers and modification of the IF model, mitigates this accuracy gap, and ensures the SNN computation is identical to the activation-quantized ANN computation. This leads to zero conversion error from the quantized ANNs, and **our SNNs achieve identical accuracy with the SOTA quantized ANNs**.
>
> Thus, this work proposes novel conversion techniques that, for the first time to our best knowledge, enable SNNs to achieve SOTA test accuracy similar to bit-serial quantized networks with significantly lower energy consumption (with neuromorphic hardware support) compared to them.
>
>
> [1] J. Wang et al., "A 28-nm Compute SRAM With Bit-Serial Logic/Arithmetic Operations for Programmable In-Memory Vector Computing," in IEEE Journal of Solid-State Circuits, vol. 55, no. 1, pp. 76-86, Jan. 2020.
>
> [2] C. Jo and K. Lee, "Bit-Serial multiplier based Neural Processing Element with Approximate adder tree," 2020 International SoC Design Conference (ISOCC), Yeosu, Korea (South), 2020, pp. 286-287.

---

> ### Author Response · Authors · 2023-11-22
> **Last day of discussion period**
>
> Dear Reviewer bagt,
>
> Thanks again for your insightful review. As the author-reviewer discussion period is about to end (Nov 22, end-of-day AoE time), we kindly request you to take a look at our response (and the updated version of the paper which incorporates your suggestions) and let us know if your concerns are addressed or if you have any follow-up questions.
>
> Thanks.

---

### Official Review · Reviewer_nh19 · 2023-11-01

**Soundness:** 3 good
**Presentation:** 3 good
**Contribution:** 3 good
**Rating:** 6
**Confidence:** 2

**Summary:**

This paper explores the source of conversion error in ANN to SNN and proposed an accurate and efficuent conversion method. The error reduction is a step-up over previous methods in identifying some extra mitigation methods that are shown to be effective as experimental results suggests

**Strengths:**

* This paper is well organized and mostly easy to follow.
* Experiments are conducted extensively in numerous dataset and compared against recent methods.
* Background are discussed throughly.
* Extra error gaps and mitigation methods, are summurized clearly over previous methods.
* Experimental results, expecially at two steps, demonstrates clear improvement in speed and accuracy over previous methods.

**Weaknesses:**

* My understanding is that this address some of the error gaps uncovered in the QCFS paper. While the paper has demnstrated faster conversion steps and better accuracy performance, would it be possible to compare and discuss energy against at least QCFS method?

**Questions:**

Please see above

---

> ### Author Response · Authors · 2023-11-17
> **Response to Reviewer nh19**
>
> Thanks for your positive comments and follow-up questions. Please see our response below.
>
>
> **My understanding is that this address some of the error gaps uncovered in the QCFS paper.**
>
> Your understanding is absolutely correct. This work addresses an error gap, in particular the deviation error, that exists in the QCFS training method. This error is the key bottleneck in ultra-low time steps that we mitigate in this work.
>
> **Comparison and Discussion of Energy against QCFS training method**
>
> We have shown the comparison of spiking activity between our and other training methods, including QCFS, and BOS in Fig. 4(b) and 4(c). Compared to these methods, our approach decreases the number of time steps and uses the bit-level regularizer, both of which reduces the spiking activity. This reduced spiking activity (almost) proportionately reduces the compute energy, since the energy overhead due to the sparsity, which involves checking whether the binary activation is zero, is typically negligible compared to the spike accumulate operation. Our approach can significantly reduce the total energy by 3.73-10.7x compared to these methods for different architectures and datasets. Note that for fair comparison, we report the energies of each method at iso-accuracy.

---

> > ### Author Response · Authors · 2023-11-22
> > **Last day of discussion period**
> >
> > Dear Reviewer nh19,
> >
> > Thanks again for your insightful review. As the author-reviewer discussion period is about to end (Nov 22, end-of-day AoE time), we kindly request you to take a look at our response (and the updated version of the paper which incorporates your suggestions) and let us know if your concerns are addressed or if you have any follow-up questions.
> >
> > Thanks.

---

### Comment · Area_Chair_WTjz · 2023-11-21
**Reminder to reviewers to participate in the author/reviewer discussion**

Dear reviewers, this is a reminder that the author/reviewer discussion period ends November 22.

This discussion is indeed supposed to be a dialog, so please respond to the comments from the authors.

@Reviewer JeWJ - thank you for already having done this!

AC